# Design Editing for Offline Model-based Optimization

**Ye Yuan**[*][†]                                                                                      *ye.yuan3@mail.mcgill.ca*
*McGill University, Mila - Quebec AI Institute*

**Youyuan Zhang**[*]                                                                              *youyuan.zhang@mail.mcgill.ca*
*McGill University*

**Can (Sam) Chen**                                                                                    *can.chen@mila.quebec*
*McGill University, Mila - Quebec AI Institute*

**Haolun Wu**                                                                                        *haolun.wu@mail.mcgill.ca*
*McGill University, Mila - Quebec AI Institute*

**Zixuan (Melody) Li**                                                                              *zixuan.li3@mail.mcgill.ca*
*McGill University, Mila - Quebec AI Institute*

**Jianmo Li**                                                                                        *jianmo.li@mail.mcgill.ca*
*McGill University*

**James J. Clark**                                                                                  *james.clark1@mcgill.ca*
*McGill University, Mila - Quebec AI Institute*

**Xue Liu**                                                                                            *xueliu@cs.mcgill.ca*
*McGill University, Mila - Quebec AI Institute*

**Reviewed on OpenReview:** *https://openreview.net/forum?id=OPFnpl7KiF*

## Abstract

Offline model-based optimization (MBO) aims to maximize a black-box objective function using only an offline dataset of designs and scores. These tasks span various domains, such as robotics, material design, and protein and molecular engineering. A common approach involves training a surrogate model using existing designs and their corresponding scores, and then generating new designs through gradient-based updates with respect to the surrogate model. This method suffers from the out-of-distribution issue, where the surrogate model may erroneously predict high scores for unseen designs. To address this challenge, we introduce a novel method, ***D**esign **E**diting for Offline **M**odel-based **O**ptimization* (**DEMO**), which leverages a diffusion prior to calibrate overly optimized designs. DEMO first generates pseudo design candidates by performing gradient ascent with respect to a surrogate model. While these pseudo design candidates contain information beyond the offline dataset, they might be invalid or have erroneously high predicted scores. Therefore, to address this challenge while utilizing the information provided by pseudo design candidates, we propose an editing process to refine these pseudo design candidates. We introduce noise to the pseudo design candidates and subsequently denoise them with a diffusion prior trained on the offline dataset, ensuring they align with the distribution of valid designs. Empirical evaluations on seven offline MBO tasks show that, with properly tuned hyperparameters, DEMO's score is competitive with the best previously reported scores in the literature. The source code is provided here.

---

[*]Equal contribution with random order.
[†]Corresponding author.

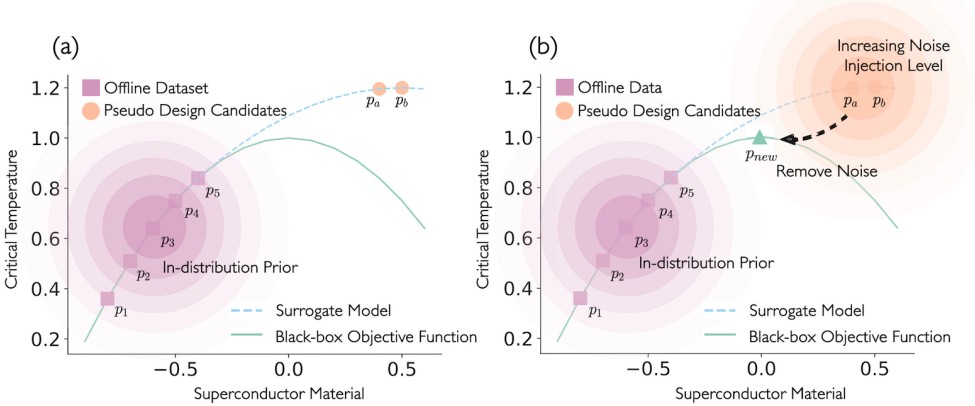

Figure 1: Illustration of DEMO: A diffusion model, acting as the prior distribution, is trained on the offline dataset. Pseudo design candidates are acquired by performing gradient ascent with respect to a learned surrogate model. New designs are generated by modifying pseudo design candidates toward the valid distribution captured by the diffusion prior.

# 1 Introduction

Designing objects with specific desired traits is a primary goal in many fields, spanning areas such as robotics, material design, and protein and molecular engineering (Trabucco et al., 2022; Liao et al., 2019; Sarkisyan et al., 2016; Angermüller et al., 2020; Hamidieh, 2018). Conventionally, this goal is pursued by iteratively testing a black-box objective function that maps a design to its property score. However, this process can be costly, time-consuming, or even hazardous (Sarkisyan et al., 2016; Angermüller et al., 2020; Hamidieh, 2018; Barrera et al., 2016; Sample et al., 2019). Thus, it is more feasible to utilize an existing offline dataset of designs and their scores to find optimal solutions without further real-world testing (Trabucco et al., 2022). This approach is known as offline model-based optimization (MBO), where the objective is to identify a design that optimizes the black-box function using only the offline dataset.

Gradient ascent is commonly used to address the offline MBO challenge. For instance, as shown in Figure 1 (a), an offline dataset might consist of five pairs of superconductor materials and their corresponding critical temperature, denoted as $p_{1,2,3,4,5}$. A deep neural network (DNN) model, referred as the *surrogate model* and represented by $f_{\boldsymbol{\theta}}(\cdot)$, is trained to approximate the unknown objective function based on this dataset. Gradient ascent is then applied to existing designs with respect to the surrogate model $f_{\boldsymbol{\theta}}(\cdot)$ for generating a new design that achieves a higher score. This approach, however, encounters an out-of-distribution (OOD) problem, where the surrogate struggles to accurately predict data outside the training distribution. This mismatch between the surrogate and the true objective function, as depicted in Figure 1 (a), can result in overly optimistic scores for the new designs generated by gradient ascent (Yu et al., 2021).

To tackle this OOD issue, recent research has proposed the use of regularization techniques, either applied directly to the surrogate model (Yu et al., 2021; Trabucco et al., 2021; Fu & Levine, 2021; Qi et al., 2022a; Yuan et al., 2023; Chen et al., 2023a) or to the design under consideration (Chen et al., 2022; 2023b). These strategies improve the surrogate's robustness and generalization. However, calibrating design candidates generated by gradient ascent remains unexplored. Instead of regularizing the surrogate models, we could tailor these design candidates toward a prior distribution and avoid over optimization.

In this work, we introduce an innovative and effective approach, ***D**esign **E**diting for Offline **M**odel-based **O**ptimization* (**DEMO**) to fill this gap. Initially, a surrogate model, represented as $f_{\boldsymbol{\theta}}(\cdot)$, is trained on the offline dataset $\mathcal{D}$, and gradient ascent is applied to existing designs with respect to the surrogate model. This process generates several designs which may have wrongly high predicted scores but low ground-truth scores due to the inaccuracies of the surrogate model, and we denote them as *pseudo design candidates*. As illustrated in Figure 1 (a), the surrogate model fits the offline data $p_1$ to $p_5$, generating pseudo design candidates $p_a$ and $p_b$ through gradient ascent. While these pseudo design candidates might be overly optimized, they contain information beyond the offline dataset. We then propose the second phase to calibrate these

pseudo design candidates and align them with the in-distribution area. Specifically, a conditional diffusion model, denoted $s_\phi(\cdot)$, is trained on all existing designs along their corresponding scores within the offline dataset to characterize a manifold of valid designs, as all existing designs are valid. After that, we edit the pseudo design candidates by introducing random noise to them and employing the diffusion prior to remove the noise. Comparing to directly leveraging a generative model to craft new design candidates from pure noise, our method benefits from the pseudo design candidates, which incorporate more information than the generative model merely trained on the offline dataset. As illustrated in Figure 1 (b), after injecting noise, the distribution of pseudo design candidates (represented by the orange contour) has more overlap with the valid design distribution (represented by the purple contour). By progressively removing the noise, we gradually project these pseudo design candidates to the manifold of valid designs, as demonstrated in Figure 1 (b). A central assumption underlying our approach is that candidate designs located near the offline data manifold are less prone to being "spurious optima" of the surrogate model. In regions far from the observed data, the model must extrapolate, which often leads to unreliable predictions and artificially inflated performance estimates. Essentially, the model may identify a high-scoring design that, in reality, is merely an artifact of overfitting or model bias. By contrast, when candidate points remain close to the data manifold, they are supported by the training data and the model's predictions in these regions are more trustworthy. Therefore, by guiding our search toward these well-supported regions, we reduce the risk of selecting designs that appear optimal only due to the model's extrapolative errors, and instead focus on candidates that are both high-performing and realistically feasible. In essence, DEMO produces new designs which are first wildly optimized and then calibrated under the constraints captured by the diffusion prior. We empirically validate DEMO across different offline MBO tasks.

In summary, this paper makes three principal contributions:

- We introduce a novel method, ***Design Editing for Offline Model-based Optimization*** (**DEMO**). DEMO first performs gradient ascent with respect to a learned surrogate model and share the information to the second phase of DEMO through pseudo design candidates.

- The second phase trains a conditional diffusion model on the offline dataset as the in-distribution prior and calibrate the pseudo design candidates with this diffusion prior to generate final designs.

- Experiments on the design-bench dataset show that, with properly tuned hyperparamters, DEMO's score is competitive with the best previously reported scores in the literature.

## 2 Preliminary

### 2.1 Offline Model-based Optimization

Offline model-based optimization (MBO) addresses a range of optimization challenges with the aim of maximizing a black-box objective function based on an offline dataset. Mathematically, we define the valid design space as $\mathcal{X} = \mathbb{R}^d$, with $d$ representing the dimension of the design. Offline MBO is formulated as:

$$\boldsymbol{x}^* = \arg\max_{\boldsymbol{x} \in \mathcal{X}} f(\boldsymbol{x}), \tag{1}$$

where $f(\cdot)$ is the black-box objective function, and $\boldsymbol{x} \in \mathcal{X}$ is a potential design. For the optimization process, we utilize an offline dataset $\mathcal{D} = \{(\boldsymbol{x}_i, y_i)\}_{i=1}^N$, with $\boldsymbol{x}_i$ representing an existing design, such as a superconductor material, and $y_i$ representing the associated property score, such as the critical temperature. Usually, this optimization process outputs $K$ candidates for optimal designs, where $K$ is a small budget to test the black-box objective function. The offline MBO problem also finds applications in other areas, like robot design, as well as protein and molecule engineering.

A prevalent approach to solving offline MBO involves approximating the unknown objective function $f(\cdot)$ with a surrogate function, typically a deep neural network (DNN) $f_{\boldsymbol{\theta}}(\cdot)$, trained on the offline dataset:

$$\boldsymbol{\theta}^* = \arg\min_{\boldsymbol{\theta}} \frac{1}{N} \sum_{i=1}^N \left(f_{\boldsymbol{\theta}}(\boldsymbol{x}_i) - y_i\right)^2. \tag{2}$$

Once the surrogate is trained, design optimization is performed using gradient ascent:

$$\boldsymbol{x}_t = \boldsymbol{x}_{t-1} + \eta \nabla_{\boldsymbol{x}} f_{\boldsymbol{\theta}}(\boldsymbol{x})\Big|_{\boldsymbol{x}=\boldsymbol{x}_{t-1}}, \quad \text{for } t \in [1, T]. \tag{3}$$

Here, $T$ represents the number of steps, and $\eta$ denotes the learning rate. The optimal design $\boldsymbol{x}^*$ is identified as $\boldsymbol{x}_T$. This gradient ascent method faces an *out-of-distribution (OOD) issue*, where the surrogate may fail to accurately predict the scores of unseen designs, resulting in suboptimal solutions.

## 2.2 Diffusion Models

Diffusion models stand out in the family of generative models due to their unique approach involving forward diffusion and backward denoising processes. The essence of diffusion models is to gradually add noise to a sample, followed by training a neural network to reverse this noise addition, thus recovering the original data distribution. In this work, we follow the formulation of diffusion models with continuous time (Song et al., 2021b; Huang et al., 2021). Here, $\boldsymbol{x}(t)$ is a random variable denoting the state of a data point at time $t \in [0, T]$. The diffusion process is defined by a stochastic differential equation (SDE):

$$\mathrm{d}\boldsymbol{x} = \boldsymbol{f}(\boldsymbol{x}, t)\mathrm{d}t + g(t)\mathrm{d}\boldsymbol{w}, \tag{4}$$

where $\boldsymbol{f}(\cdot, t)$ is the drift coefficient, $g(\cdot)$ is the diffusion coefficient, and $\boldsymbol{w}$ is a standard Wiener process. The backward denoising process is given by the reverse time SDE:

$$\mathrm{d}\boldsymbol{x} = \left[\boldsymbol{f}(\boldsymbol{x}, t) - g(t)^2 \nabla_{\boldsymbol{x}} \log p_t(\boldsymbol{x})\right] \mathrm{d}t + g(t)\,\mathrm{d}\bar{\boldsymbol{w}}, \tag{5}$$

where $\mathrm{d}t$ represents a negative infinitesimal step in time, and $\bar{w}$ is a reverse time Wiener process. The gradient of the log probability, $\nabla_{\boldsymbol{x}} \log p_t(\boldsymbol{x})$, is approximated by a neural network $s_{\boldsymbol{\phi}}(\boldsymbol{x}(t), t)$ with score-matching objectives (Vincent, 2011; Song & Ermon, 2019).

Beyond basic diffusion models, our focus is to train a conditional diffusion model that learns the conditional probability distribution of designs based on their associated property scores. To incorporate conditions to diffusion models, Ho & Salimans (2022) achieve it by dividing the score function into a combination of conditional and unconditional components, known as classifier-free diffusion models. Specifically, a single neural network, $s_{\boldsymbol{\phi}}(\boldsymbol{x}_t, t, y)$, is trained to handle both components by utilizing $y$ as the condition or leaving it empty for unconditional functions. Formally, we can write this combination as follows:

$$s_{\boldsymbol{\phi}}(\boldsymbol{x}_t, t, y) = (1 + \omega)s_{\boldsymbol{\phi}}(\boldsymbol{x}_t, t, y) - \omega s_{\boldsymbol{\phi}}(\boldsymbol{x}_t, t), \tag{6}$$

where $\omega$ is a parameter that adjusts the influence of the conditions. A higher value of $\omega$ ensures that the generation process adheres more closely to the specified conditions, while a lower $\omega$ value allows greater flexibility in the outputs.

# 3 Related Works

## 3.1 Offline Model-based Optimization

Recent offline model-based optimization (MBO) techniques broadly fall into two categories: *(i)* those that employ gradient-based optimizations and *(ii)* those that create new designs via generative models. Gradient-based methods directly optimize the surrogate model's predictions using gradient-based search. However, straightforward gradient ascent can lead to candidate designs that are far from the observed data, where the surrogate is prone to unreliable extrapolations. Therefore, to address this problem several works employ regularization techniques that enhance either the surrogate model (Yu et al., 2021; Trabucco et al., 2021; Fu & Levine, 2021; Qi et al., 2022a) or the design itself (Chen et al., 2023b; 2022), thus improving the model's robustness and generalization capacity. BOSS (Dao et al., 2024b) introduces a sensitivity-informed regularizer to mitigate the overfitting and narrow prediction margins of offline surrogates. IGNITE (Dao et al., 2024a) presents a model-agnostic sharpness regularization method that theoretically reduces the generalization error of offline surrogate models. aSCR (Yao et al., 2024) constrains the optimization trajectory to regions

where the surrogate is reliable by dynamically adjusting regularization strength. Some approaches involve synthesizing new data with pseudo labels (Yuan et al., 2023; Chen et al., 2023a), where they aim to identify useful information from these synthetic data to correct the surrogate model's inaccuracies. Other approaches like PGS (Chemingui et al., 2024) employs policy-guided gradient search approach that explicitly learns the best policy for a given surrogate model. Match-OPT (Hoang et al., 2024) provides a theoretical framework for offline black-box optimization by quantifying the performance gap due to surrogate inaccuracies and introduces a black-box gradient matching algorithm to improve surrogate quality. The second category encompasses methods that learn to replicate the conditional distribution of existing designs and their scores, including approaches such as MIN (Kumar & Levine, 2020), CbAS (Brookes et al., 2019), Auto CbAS (Fannjiang & Listgarten, 2020), DDOM (Krishnamoorthy et al., 2023), and BONET (Mashkaria et al., 2023). ExPT (Nguyen et al., 2023) trains a foundation model for few-shot experimental design that leverages unsupervised pretraining and in-context learning on unlabeled data to efficiently generate candidate optima with minimal labeled examples. A concurrent work (Dao et al., 2025) unsurprisingly shares the core idea of leveraging diffusion processes to bridge the gap between low-and-high performing designs. This work explicitly trains a generalized diffusion process to map directly between the distributions of low- and high-value designs. In contrast, we leverage a diffusion prior in a two-phase process by first generating pseudo design candidates via surrogate-based gradient ascent, and then employing a diffusion model to edit these candidates so that they remain on the valid design manifold. Our decoupled two optimization phases and modularity provide enhanced stability and flexibility. These methods are known for their ability to generate designs by sampling from learned distributions conditioned on higher target scores.

In DEMO, a candidate design is first generated via gradient-based optimization, ensuring that the candidate is high-scoring according to the surrogate model. Recognizing that direct gradient-based updates can push the candidate into regions where the model's predictions are unreliable, we then employ a diffusion model to "edit" the design, explicitly moving it closer to the data manifold. This two-stage process decouples the search for high performance from the regularization required to maintain realism. While traditional gradient-based methods embed regularization within the optimization loop and generative methods rely solely on learned distributions, DEMO leverages the flexibility of gradient-based search alongside the robustness of diffusion-based editing. This approach effectively reduces the risk of spurious optima-candidates that appear optimal due to extrapolative errors, by ensuring that final designs remain in regions well-supported by data. In summary, while existing offline MBO methods typically focus on either direct gradient-based optimization with embedded regularization or on generative modeling of the design space, DEMO's hybrid approach offers a novel balance between performance and realism, thereby addressing key challenges inherent in both paradigms.

## 3.2 Diffusion-Based Editing

Diffusion models have shown remarkable success in various generation tasks across multiple modalities, especially for their ability to control the generation process based on given conditions. For instance, recent advancements have utilized diffusion models for zero-shot, test-time editing in the domains of text-based image and video generation. SDEdit (Meng et al., 2022) employs an editing strategy to balance realism and faithfulness in image generation. To improve the reconstruction quality, methodologies such as DDIM Inversion (Song et al., 2021a), Null-text Inversion (Mokady et al., 2023) and Negative-prompt Inversion (Miyake et al., 2023) concentrate on deterministic mappings from source latents to initial noise, conditioned on source text. Building on these, CycleDiffusion (Wu & la Torre, 2023) and Direct Inversion (Ju et al., 2023) leverage source latents from each inversion step and further improve the faithfulness of the target image to the source image. Following the image editing technique, several video editing methods (Qi et al., 2023; Ceylan et al., 2023; Yang et al., 2023; Geyer et al., 2024; Cong et al., 2024; Zhang et al., 2024) adopt image diffusion models and enforce temporal consistency across frames, offering practical and efficient solutions for video editing. Inspired by the success of these editing techniques in the field of computer vision, DEMO distinguishes itself in the context of offline MBO by first leveraging the surrogate model to craft pseudo design candidates and then refining them toward the in-distribution area.

# 4 Methodology

In this section, we elaborate on the details of our proposed ***D**esign **E**diting for Offline **M**odel-based **O**ptimization* (**DEMO**). Typically, the out-of-distribution (OOD) issue in offline MBO arises due to incomplete observation of the entire distribution of designs and scores. During gradient ascent optimization, the pseudo design candidates are optimized with respect to the surrogate model without constraints, which leads them into the OOD region and causes overestimation of the scores. To address this, we introduce a design editing process to calibrate the pseudo design candidates toward the in-distribution area, which mitigates the OOD problem through the use of a learned diffusion prior. Algorithm 1 illustrates the complete process of DEMO.

## 4.1 Acquirement of Pseudo Design Candidates

Initially, a deep neural network (DNN), denoted as $f_{\boldsymbol{\theta}}(\cdot)$ with parameters $\boldsymbol{\theta}$, is trained on the offline dataset $\mathcal{D} = \{(\boldsymbol{x}_i, y_i)\}_{i=1}^N$, where $\boldsymbol{x}_i$ and $y_i$ denote a design and its associated score, respectively. The parameters $\boldsymbol{\theta}$ are optimized as shown in Eq. (2). The solution $f_{\boldsymbol{\theta}^*}(\cdot)$ obtained from Eq. (2) serves as a surrogate for the unknown black-box objective function $f(\cdot)$ in Eq. (1). New data are then generated by performing gradient ascent on the existing designs with respect to the learned surrogate model $f_{\boldsymbol{\theta}^*}(\cdot)$. The initial point $\boldsymbol{x}_0$ is an existing design selected from $\mathcal{D}$. We update it as shown in Eq. (3). The design $\boldsymbol{x}_T$ acquired at step $T$ is an overly optimized design, as no constraints are applied during the optimization process. By iteratively using the top $K$ designs in the offline dataset $\mathcal{D}$ as the initial points, a batch of pseudo design candidates, denoted as $\mathcal{D}'$, is acquired. This process is outlined from line 2 to line 8 in Algorithm 1.

## 4.2 Training of Diffusion Prior

We employ a classifier-free conditional diffusion model (Ho & Salimans, 2022) to learn the conditional probability distribution of existing designs and their scores in offline dataset $\mathcal{D}$. Following the approach in DDOM (Krishnamoorthy et al., 2023), we use the Variance Preserving (VP) stochastic differential equation (SDE) for the forward diffusion process, as specified in Song et al. (2021b):

$$\mathrm{d}\boldsymbol{x} = -\frac{\beta(t)}{2}\boldsymbol{x}\mathrm{d}t + \sqrt{\beta(t)}\mathrm{d}\boldsymbol{w}, \tag{7}$$

where $\beta(t)$ is a continuous time function for $t \in [0, 1]$. The forward process in DDPM (Ho et al., 2020) is proved to be a discretization of Eq. (7) (Song et al., 2021b). To integrate conditions in the backward denoising process, we need to train a DNN $s_{\boldsymbol{\phi}}(\boldsymbol{x}_t, t, y)$ with parameters $\boldsymbol{\phi}$, conditioned on the time $t$ and the score $y$ associated with the unperturbed design $\boldsymbol{x}_0$ corresponding to $\boldsymbol{x}_t$. The parameters $\boldsymbol{\phi}$ are optimized as:

$$\boldsymbol{\phi}^* = \arg\min_{\boldsymbol{\phi}} \mathbb{E}_t \left[\lambda(t)\mathbb{E}_{\boldsymbol{x}_0, y} \left[\mathbb{E}_{\boldsymbol{x}_t|\boldsymbol{x}_0} \left[\|\boldsymbol{s}_{\boldsymbol{\phi}}(\boldsymbol{x}_t, t, y) - \nabla_{\boldsymbol{x}} \log p_t(\boldsymbol{x}_t|\boldsymbol{x}_0)\|^2\right]\right]\right], \tag{8}$$

where $\lambda(t)$ is a positive weighting function depending on time. Since we train on the offline dataset $\mathcal{D}$, the model optimized according to Eq. (8) captures the manifold of valid designs. This part is described in Line 10 of Algorithm 1.

However, since the offline dataset may contain very low-score existing designs, naively applying the trained model to the subsequent design editing process might be harmful for calibrating the pseudo design candidates. Therefore, we propose to use the distribution conditioned on the maximum property score among the offline dataset as the prior for later usage, which is a simple yet effective method for mitigating the negative impacts of very low-score designs.

## 4.3 Design Editing Process

Due to potential inaccuracies of the surrogate model $f_{\boldsymbol{\theta}^*}(\cdot)$ in representing the black-box objective function, the set of pseudo design candidates $\mathcal{D}'$ might include samples that have high predicted scores but low ground-truth scores. Inspired by the success of editing techniques in image synthesis tasks (Meng et al., 2022; Su et al., 2023), we explore the potential of calibrating these pseudo design candidates to obtain new designs

with valid high scores. We perturb a pseudo design candidate $\boldsymbol{x}^{(p)} \in \mathcal{D}'$ by introducing noise at a specific time $m$ out of $\{1, \cdots, M\}$ and auxiliary noise levels $\beta_1, \cdots, \beta_M$:

$$\boldsymbol{x}^{(p)}_{\text{perturb}} = \sqrt{\bar{\alpha}_m}\boldsymbol{x}^{(p)} + \sqrt{1 - \bar{\alpha}_m}\epsilon, \tag{9}$$

where $\alpha_m = 1 - \beta_m$, $\bar{\alpha}_m = \prod_{s=1}^m \alpha_s$, and $\epsilon \sim \mathcal{N}(\mathbf{0}, \mathbf{I})$. This results in a closed-form expression that samples $\boldsymbol{x}^{(p)}_{\text{perturb}} \sim \mathcal{N}(\sqrt{\bar{\alpha}_m}\boldsymbol{x}^{(p)}, (1 - \bar{\alpha}_m)\mathbf{I})$. The perturbed design is then used as the starting point. A final optimized design is synthesized by using any numerical solver for the backward denoising process with the model $s_{\boldsymbol{\phi}^*}(\cdot)$, conditioned on the maximum property score among the offline dataset, to remove the noise. In our implementation, we follow existing studies (Krishnamoorthy et al., 2023) and use Heun's method as the solver. To yield $K$ final optimized designs, we utilize all pseudo design candidates from $\mathcal{D}'$, obtain various perturbed designs, and denoise them, pushing them toward the prior distribution. Lines 12 to 17 of Algorithm 1 present the process of this procedure.

---

**Algorithm 1** Design Editing for Offline Model-based Optimization

**Input:** Offline dataset $\mathcal{D} = \{(\boldsymbol{x}_i, y_i)\}_{i=1}^N$, and a time $m$.
**Output:** $K$ candidate optimal designs.
1: /* Acquirement of Pseudo Design Candidates */
2: Initialize a surrogate model $f_{\boldsymbol{\theta}}(\cdot)$ and optimize $\boldsymbol{\theta}$ with Eq. (2) to obtain $f_{\boldsymbol{\theta}^*}(\cdot)$.
3: $\mathcal{D}' = \{\}$
4: **for** $i = 1, 2, \cdots, K$ **do**
5: $\quad$ $\boldsymbol{x}_0 \longleftarrow \boldsymbol{x}_i$ with the $i$-th best score within $\mathcal{D}$.
6: $\quad$ **for** $t = 1, 2, \cdots, T$ **do**
7: $\quad\quad$ Update $\boldsymbol{x}_t$ with Eq. (3).
8: $\quad$ Append $\boldsymbol{x}_T$ to $\mathcal{D}'$.
9: /* Training of Diffusion Prior */
10: Initialize $s_{\boldsymbol{\phi}}(\cdot)$ and optimize $\boldsymbol{\phi}$ with Eq. (8) on $\mathcal{D}$ to obtain $s_{\boldsymbol{\phi}^*}(\cdot)$.
11: /* Design Editing Process */
12: Candidates = $\{\}$
13: **for** $i = 1, 2, \cdots, K$ **do**
14: $\quad$ $\boldsymbol{x}^{(p)} \longleftarrow \boldsymbol{x}_i \in \mathcal{D}'$
15: $\quad$ Perturb $\boldsymbol{x}^{(p)}$ with Eq. (9) and the given time $m$.
16: $\quad$ Denoise $\boldsymbol{x}^{(p)}_{perturb}$ and generate $\boldsymbol{x}_{new}$ using the Heun's method with $s_{\boldsymbol{\phi}^*}(\cdot)$ conditioned on $\max(\{y_i\}_{i=1}^N)$.
17: $\quad$ Append $\boldsymbol{x}_{new}$ to Candidates.
18: **return** Candidates

---

## 5 Experiments

This section first describes the experiment setup, followed by the implementation details and results. We aim to answer the following questions in this section: *(Q1)* Is our proposed DEMO more effective than baseline methods in addressing the offline MBO problem? *(Q2)* One can alternatively generate new designs from the diffusion prior or directly use the pseudo design candidates. Is DEMO better than these partial alternatives by introducing the editing process?

### 5.1 Dataset and Tasks

We carry out experiments on 7 tasks selected from Design-Bench (Trabucco et al., 2022) and BayesO Benchmarks (Kim, 2023), including 4 continuous tasks and 3 discrete tasks. The continuous tasks are as follows: *(i)* Superconductor (SuperC) (Hamidieh, 2018), where the goal is to create a superconductor with 86 continuous components to maximize critical temperature, using $17,014$ designs; *(ii)* Ant Morphology (Ant) (Trabucco et al., 2022; Brockman et al., 2016), where the objective is to design a four-legged ant with 60 continuous components to increase crawling speed, based on $10,004$ designs; *(iii)* D'Kitty Morphology

(D'Kitty) (Trabucco et al., 2022; Ahn et al., 2020), where the focus is on designing a four-legged D'Kitty with 56 continuous components to enhance crawling speed, using $10,004$ designs; *(iv)* Inverse Levy Function (Levy) (Kim, 2023), where the aim is to maximize function values of the inverse black-box Levy function with 60 input dimensions, using $15,000$ designs. The discrete tasks include: *(v)* TF Bind 8 (TF8) (Barrera et al., 2016), where the goal is to identify an 8-unit DNA sequence that maximizes binding activity score, with $32,898$ designs; *(vi)* TF Bind 10 (TF10) (Barrera et al., 2016), where the objective is to find a 10-unit DNA sequence that optimizes binding activity score, using $30,000$ designs; *(vii)* NAS (Zoph & Le, 2017), where the aim is to discover the optimal neural network architecture to improve test accuracy on the CIFAR-10 dataset (Hinton et al., 2012), using $1,771$ designs.

## 5.2 Evaluation and Metrics

Following the evaluation protocol used in previous studies (Trabucco et al., 2022), we assume the budget $K = 128$ and generate 128 new designs for each method. The 100-th (max) percentile normalized ground-truth score is reported in this section, and the 50-th (median) percentile score is provided in Appendix A.1. This normalized score is calculated as $y_n = \frac{y - y_{\min}}{y_{\max} - y_{\min}}$, where $y_{\min}$ and $y_{\max}$ are the minimum and maximum scores in the entire offline dataset, respectively. For better comparison, we include the normalized score of the best design in the offline dataset, denoted as $\mathcal{D}(\mathbf{best})$. Additionally, we provide mean and median rankings across all 7 tasks for a comprehensive performance evaluation.

## 5.3 Comparison Methods

We benchmark DEMO against three groups of baseline approaches: *(i)* traditional methods, *(ii)* those utilizing gradient optimizations from current designs, and *(iii)* those employing generative models for sampling. Traditional methods include: *(1)* **BO-qEI** (Wilson et al., 2017): conducts Bayesian Optimization to maximize the surrogate, proposes designs using the quasi-Expected-Improvement acquisition function, and labels the designs using the surrogate model. *(2)* **CMA-ES** (Hansen, 2006): progressively adjusts the distribution toward the optimal design by altering the covariance matrix. *(3)* **REINFORCE** (Williams, 1992): optimizes the distribution over the input space using the learned surrogate. The second category includes: *(4)* **Mean**: optimizes the average prediction of the ensemble of surrogate models. *(5)* **Min**: optimizes the lowest prediction from a group of learned objective functions. *(6)* **COMs** (Trabucco et al., 2021): applies regularization to assign lower scores to designs derived through gradient ascent. *(7)* **ROMA** (Yu et al., 2021): introduces smoothness regularization to the DNN. *(8)* **NEMO** (Fu & Levine, 2021): limits the discrepancy between the surrogate and the black-box objective function using normalized maximum likelihood before performing gradient ascent. *(9)* **BDI** (Chen et al., 2022) employs forward and backward mappings to transfer knowledge from the offline dataset to the design. *(10)* **IOM** (Qi et al., 2022b): ensures representation consistency between the training dataset and the optimized designs. *(11)* **ICT** (Yuan et al., 2023): identifies useful information from a pseudo-labeled dataset to improve the surrogate model. *(12)* **Tri-mentoring** (Chen et al., 2023a): leverages information of pairwise comparison data to enhance ensemble performance. *(13)* **PGS** (Chemingui et al., 2024): learns the best policy for a given surrogate model. Generative model-based methods include: *(14)* **CbAS** (Brookes et al., 2019), which adapts a VAE model to steer the design distribution toward areas with higher scores. *(15)* **Auto CbAS** (Fannjiang & Listgarten, 2020), which uses importance sampling to update a regression model based on CbAS. *(16)* **MIN** (Kumar & Levine, 2020), which establishes a relationship between scores and designs and seeks optimal designs within this framework. *(17)* **DDOM** (Krishnamoorthy et al., 2023), which learns a generative diffusion model conditioned on the score values. *(18)* **BONET** (Mashkaria et al., 2023), which employs an autoregressive model trained on the offline dataset.

## 5.4 Implementation Details

We follow the training protocols from Trabucco et al. (2021) for all comparative methods unless stated otherwise. A 3-layer MLP with ReLU activation is used for both $f_{\boldsymbol{\theta}}(\cdot)$ and $s_{\boldsymbol{\phi}}(\cdot)$, with a hidden layer size of 2048. In Algorithm 1, the iteration count, $T$, is established at 100 for both continuous and discrete tasks. The Adam optimizer (Kingma & Ba, 2015) is utilized to train the surrogate models over 200 epochs with a batch

Table 1: Experimental results on continuous tasks for comparison.

| Method | Superconductor | Ant Morphology | D'Kitty Morphology | Levy |
|---|---|---|---|---|
| $\mathcal{D}(\mathbf{best})$ | 0.399 | 0.565 | 0.884 | 0.613 |
| BO-qEI | $0.402 \pm 0.034$ | $0.819 \pm 0.000$ | $0.896 \pm 0.000$ | $0.810 \pm 0.016$ |
| CMA-ES | $0.465 \pm 0.024$ | $\mathbf{1.214 \pm 0.732}$ | $0.724 \pm 0.001$ | $0.887 \pm 0.025$ |
| REINFORCE | $0.481 \pm 0.013$ | $0.266 \pm 0.032$ | $0.562 \pm 0.196$ | $0.564 \pm 0.090$ |
| Mean | $0.505 \pm 0.013$ | $0.940 \pm 0.014$ | $\mathbf{0.956 \pm 0.014}$ | $\mathbf{0.984 \pm 0.023}$ |
| Min | $0.501 \pm 0.019$ | $0.918 \pm 0.034$ | $0.942 \pm 0.009$ | $0.964 \pm 0.023$ |
| COMs | $0.481 \pm 0.028$ | $0.842 \pm 0.037$ | $0.926 \pm 0.019$ | $0.936 \pm 0.025$ |
| ROMA | $0.509 \pm 0.015$ | $0.916 \pm 0.030$ | $0.929 \pm 0.013$ | $0.976 \pm 0.019$ |
| NEMO | $0.502 \pm 0.002$ | $0.955 \pm 0.006$ | $0.952 \pm 0.004$ | $0.969 \pm 0.019$ |
| BDI | $0.513 \pm 0.000$ | $0.906 \pm 0.000$ | $0.919 \pm 0.000$ | $0.938 \pm 0.000$ |
| IOM | $\mathbf{0.518 \pm 0.020}$ | $0.922 \pm 0.030$ | $0.944 \pm 0.012$ | $\mathbf{0.988 \pm 0.021}$ |
| ICT | $0.503 \pm 0.017$ | $\mathbf{0.961 \pm 0.007}$ | $\mathbf{0.968 \pm 0.020}$ | $0.879 \pm 0.018$ |
| Tri-mentoring | $\mathbf{0.514 \pm 0.018}$ | $0.948 \pm 0.014$ | $\mathbf{0.966 \pm 0.010}$ | $0.924 \pm 0.035$ |
| PGS | $\mathbf{0.563 \pm 0.058}$ | $0.949 \pm 0.017$ | $\mathbf{0.966 \pm 0.013}$ | $0.963 \pm 0.027$ |
| CbAS | $\mathbf{0.503 \pm 0.069}$ | $0.876 \pm 0.031$ | $0.892 \pm 0.008$ | $0.938 \pm 0.037$ |
| Auto CbAS | $0.421 \pm 0.045$ | $0.882 \pm 0.045$ | $0.906 \pm 0.006$ | $0.797 \pm 0.033$ |
| MIN | $0.499 \pm 0.017$ | $0.445 \pm 0.080$ | $0.892 \pm 0.011$ | $0.761 \pm 0.037$ |
| DDOM | $0.486 \pm 0.013$ | $0.952 \pm 0.007$ | $0.941 \pm 0.006$ | $0.927 \pm 0.031$ |
| BONET | $0.437 \pm 0.022$ | $\mathbf{0.976 \pm 0.012}$ | $0.954 \pm 0.012$ | $0.918 \pm 0.025$ |
| $\mathbf{DEMO}_{(ours)}$ | $\mathbf{0.525 \pm 0.009}$ | $\mathbf{0.968 \pm 0.009}$ | $\mathbf{0.970 \pm 0.007}$ | $\mathbf{1.007 \pm 0.015}$ |

size of 128, and a learning rate set at $10^{-1}$. The step size, $\eta$, in Eq. (3) is configured at $10^{-3}$ for continuous tasks and $10^{-1}$ for discrete tasks. The diffusion model, $s_\phi(\cdot)$, undergoes training for 200 epochs with a batch size of 128. Previous works such as Krishnamoorthy et al. (2023) employ 1000 epochs for training diffusion models, but we find 200 epochs is enough for the diffusion models to converge. For the *design editing process*, following precedents set by previous studies (Krishnamoorthy et al., 2023), we set $M$ at 1000. The selected value of $m$ is 600, with further elaboration provided in Appendix A.2. Results from traditional methodologies are referenced from Trabucco et al. (2022), and we conduct 8 independent trials for other methods, reporting the mean and standard error. All experiments are conducted on a workstation with a single Intel Xeon Platinum 8160T CPU and a single NVIDIA Tesla V100 GPU, with execution times per trial ranging from 10 minutes to 20 hours (including evaluation time), depending on the specific tasks.

## 5.5   Results

Following existing studies (Trabucco et al., 2021; Yuan et al., 2023), in Table 1 and Table 2 we mark a method in bold if its mean is at least as high as the highest mean minus one standard deviation of the corresponding method. Alternatively, a method is also bolded if its mean plus one standard deviation reaches or exceeds the method with the highest mean.

**Performance in Continuous Tasks.** Table 1 presents the results of the four continuous tasks. DEMO achieves competitive performance to existing approaches across all of continuous tasks. DEMO outperforms gradient-based methods, such as NEMO and ICT, by leveraging the design editing process to calibrate the pseudo design candidates rather than performing unconstrained optimization against regularized surrogate models. It is worth noting that some methods, such as COMs, employ a constrained optimization process by penalizing the value at a lookahead gradient ascent optimization point. The superior performance of DEMO compared to COMs indicates that the design editing process is a more effective method. Moreover, when compared to other generative model-based approaches, such as MIN and DDOM, DEMO generally outperforms them because these methods train models solely on the offline dataset and may not benefit from the information provided by surrogate models. DEMO achieves better performance by effectively utilizing the surrogate model to acquire a batch of pseudo design candidates. These results strongly support the effectiveness of DEMO for continuous tasks.

Table 2: Experimental results on discrete tasks, and ranking on all tasks for comparison.

| Method | TF Bind 8 | TF Bind 10 | NAS | Rank Mean | Rank Median |
|---|---|---|---|---|---|
| $\mathcal{D}(\textbf{best})$ | 0.439 | 0.467 | 0.436 | | |
| BO-qEI | $0.798 \pm 0.083$ | $0.652 \pm 0.038$ | $\textbf{1.079} \pm \textbf{0.059}$ | 13.9/19 | 16/19 |
| CMA-ES | $0.953 \pm 0.022$ | $0.670 \pm 0.023$ | $0.985 \pm 0.079$ | 9.1/19 | 7/19 |
| REINFORCE | $0.948 \pm 0.028$ | $0.663 \pm 0.034$ | $-1.895 \pm 0.000$ | 15.1/19 | 19/19 |
| Mean | $0.895 \pm 0.020$ | $0.654 \pm 0.028$ | $0.663 \pm 0.058$ | 9.3/19 | 9/19 |
| Min | $0.931 \pm 0.036$ | $0.634 \pm 0.033$ | $0.708 \pm 0.027$ | 10.7/19 | 11/19 |
| COMs | $0.474 \pm 0.053$ | $0.625 \pm 0.010$ | $0.796 \pm 0.029$ | 13.1/19 | 14/19 |
| ROMA | $0.921 \pm 0.040$ | $0.669 \pm 0.035$ | $0.934 \pm 0.025$ | 7.9/19 | 7/19 |
| NEMO | $0.942 \pm 0.003$ | $\textbf{0.708} \pm \textbf{0.010}$ | $0.735 \pm 0.012$ | 6.7/19 | 7/19 |
| BDI | $0.870 \pm 0.000$ | $0.605 \pm 0.000$ | $0.722 \pm 0.000$ | 12.1/19 | 13/19 |
| IOM | $0.870 \pm 0.074$ | $0.648 \pm 0.025$ | $0.411 \pm 0.044$ | 10.0/19 | 10/19 |
| ICT | $0.958 \pm 0.008$ | $0.691 \pm 0.023$ | $0.667 \pm 0.091$ | 7.7/19 | 6/19 |
| Tri-mentoring | $\textbf{0.970} \pm \textbf{0.001}$ | $\textbf{0.722} \pm \textbf{0.017}$ | $0.759 \pm 0.102$ | 5.4/19 | 4/19 |
| PGS | $\textbf{0.981} \pm \textbf{0.015}$ | $0.658 \pm 0.021$ | $0.727 \pm 0.033$ | 5.4/19 | 7/19 |
| CbAS | $0.927 \pm 0.051$ | $0.651 \pm 0.060$ | $0.683 \pm 0.079$ | 12.0/19 | 12/19 |
| Auto CbAS | $0.910 \pm 0.044$ | $0.630 \pm 0.045$ | $0.506 \pm 0.074$ | 15.6/19 | 16/19 |
| MIN | $0.905 \pm 0.052$ | $0.616 \pm 0.021$ | $0.717 \pm 0.046$ | 15.4/19 | 16/19 |
| DDOM | $0.961 \pm 0.024$ | $0.640 \pm 0.029$ | $0.737 \pm 0.014$ | 9.4/19 | 10/19 |
| BONET | $\textbf{0.975} \pm \textbf{0.004}$ | $0.681 \pm 0.035$ | $0.724 \pm 0.008$ | 8.0/19 | 6/19 |
| $\textbf{DEMO}_{(\text{ours})}$ | $\textbf{0.982} \pm \textbf{0.016}$ | $\textbf{0.762} \pm \textbf{0.058}$ | $0.753 \pm 0.017$ | $\textbf{2.1/19}$ | $\textbf{1/19}$ |

**Performance in Discrete Tasks.** Table 2 displays the results of the three discrete tasks. DEMO achieves top performance in TF Bind 8 and TF Bind 10, with the results on TF10 surpassing those of other methods by a significant margin, suggesting DEMO's capability in solving discrete offline MBO tasks. However, DEMO underperforms on NAS, which may be due to two reasons. First, each neural network architecture is encoded as a sequence of one-hot vectors, which has a length of 64. This encoding process might be insufficient for precisely representing all features of a given architecture, leading to suboptimal performance on NAS. Additionally, after examining the NAS offline dataset, we found that many existing designs share commonalities. This redundancy means that the NAS dataset contains less useful information compared to other tasks, which further explains why DEMO's performance on NAS is not as strong.

**Summary.** These results on both continuous and discrete tasks provide a clear answer to $Q1$. DEMO attains the highest rankings with a mean of 2.1/19 and a median of 1/19, as detailed in Table 2 and Figure 2b, and secures top performance in all tasks. We perform Welch's t-test across the seven tasks to compare DEMO against all baseline methods. To account for multiple hypothesis tests, we apply the Bonferroni correction to all reported p-values to control the family-wise error rate. The complete set of corrected p-values is provided in Appendix A.3. In summary, at a significance level of $\alpha = 0.05$, the results confirm that DEMO achieves statistically significant improvements over 10 baseline methods in the Superconductor task, 14 in the Ant task, 13 in the D'Kitty task, 16 in the Levy task, 11 in the TF8 task, 14 in the TF10 task, and 7 in the NAS task. We further examine the reliability of DEMO in Appendix A.4. Additional quantitative analysis of the predicted and ground-truth property scores of the pseudo design candidates and final candidates is provided in Appendix A.5. We also study the influence of the number of gradient ascent steps in the convergence study detailed in Appendix A.5.

## 5.6 Ablation Study

New designs can be alternatively generated from the prior distribution using the diffusion model or by directly using the pseudo design candidates after the gradient ascent process. We refer to these two methods as Diffusion-only and Grad-only, respectively. To rigorously assess whether the introduction of the editing phase within our DEMO method is beneficial, ablation experiments are conducted by systematically comparing the complete DEMO method with Diffusion-only and Grad-only. The results, summarized in Table 2a, provide clear insights into the impact of our design editing process. For the four continuous tasks, DEMO consistently achieves higher performance compared to its ablated versions. For instance, in the SuperC

| Task | DEMO | Grad-only | Diffusion-only |
|---|---|---|---|
| SuperC | **0.525 ± 0.009** | 0.482 ± 0.013 | 0.346 ± 0.011 |
| Ant | **0.968 ± 0.009** | 0.963 ± 0.008 | 0.377 ± 0.004 |
| D'Kitty | **0.970 ± 0.007** | 0.933 ± 0.002 | 0.762 ± 0.023 |
| Levy | **1.007 ± 0.015** | 0.990 ± 0.020 | 0.483 ± 0.015 |
| TF8 | **0.980 ± 0.004** | 0.965 ± 0.008 | 0.420 ± 0.004 |
| TF10 | **0.762 ± 0.058** | 0.638 ± 0.019 | 0.465 ± 0.006 |
| NAS | **0.753 ± 0.017** | 0.668 ± 0.084 | 0.274 ± 0.013 |

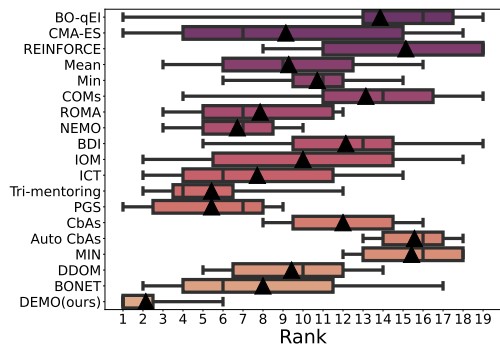

(a) Ablation studies of DEMO.

(b) Comparison of ranks across methods.

Figure 2

| Task | Dimension | Memory | Training | Optimization |
|---|---|---|---|---|
| SuperC | (17014, 86) | 1179 | 235.92 | 0.92 |
| Ant | (10004, 60) | 1177 | 220.82 | 0.82 |
| D'Kitty | (10004, 56) | 1177 | 222.68 | 0.68 |
| Levy | (15000, 60) | 1177 | 227.80 | 0.80 |
| TF8 | (32898, 8) | 1177 | 253.81 | 0.81 |
| TF10 | (30000, 10) | 1177 | 311.73 | 0.73 |
| NAS | (1771, 64) | 1201 | 233.81 | 0.81 |

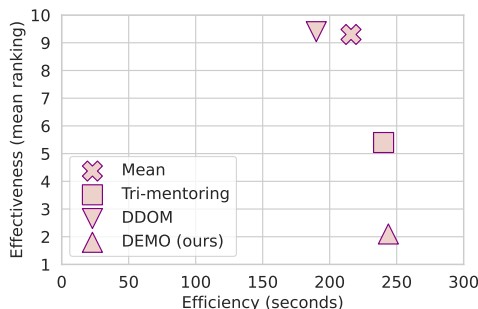

(a) Dimension records the number of samples in the offline dataset of each task and the corresponding input dimension. Memory summarizes the GPU usage in `MB` for training the models, and the last two columns demonstrate the time used for training models and optimizing candidate designs in `seconds`, respectively.

(b) Comparison of effectiveness-efficiency trade-off with other representative methods. The left-bottom corner represents that methods are both effective (with higher mean ranking) and efficient (less training and optimization latency).

Figure 3: Computational efficiency analysis.

task, DEMO achieves a score of $0.525 \pm 0.009$, significantly higher than both Grad-only ($0.482 \pm 0.013$) and Diffusion-only ($0.346 \pm 0.011$). Unlike baseline methods of conditional generative models based on a target score higher than all existing designs, such as Auto CbAS and DDOM, the Diffusion-only approach generates new designs conditioned on the maximum score within the offline dataset for the ablation study purpose. Similar improvements are observed in the Ant, D'Kitty, and Levy tasks, underscoring the effectiveness of integrating the design editing process in continuous tasks. In the discrete tasks TF8, TF10, and NAS, DEMO's superior performance over both alternative solutions is evident, highlighting its comprehensive effectiveness in managing discrete challenges. Overall, the ablation studies validate the importance of the design editing process within the DEMO method, answering *Q*2 conclusively. The complete DEMO method collectively contributes to enhancements across a range of both continuous and discrete tasks and various input dimensions.

### 5.7 Computational Efficiency Analysis

To evaluate the computational efficiency and scalability of DEMO, we run additional experiments on our workstation with a single Intel Xeon Platinum 8160T CPU and a single NVIDIA Tesla V100 GPU. The results are summarized in Table 3a. The table reports the sizes of the offline dataset, the corresponding input dimension, the GPU memory usage for training models in megabytes (MB), and the time for training models and optimizing design candidates in seconds. Furthermore, we closely examine the time efficiency for optimizing design candidates of another diffusion-based method, DDOM, on every task. DDOM takes 2.35 seconds in the Superconductor task, 2.21 seconds in the Ant task, 1.84 seconds in the D'Kitty task, 2.07

seconds in the Levy task, 1.89 seconds in the TF Bind 8 task, 1.83 seconds in the TF Bind 10 task, and 1.85 seconds in the NAS task. This indicates that our DEMO is more time-efficient than DDOM in every task for optimizing design candidates. In addition, Figure 3b illustrates the effectiveness-efficiency trade-off for DEMO compared to DDOM and two representative surrogate-based methods, Mean Ensemble and Tri-mentoring. Mean Ensemble is a widely used traditional method for offline MBO that aggregates predictions from an ensemble of surrogate models to provide robust estimates, while Tri-mentoring is a more recent baseline that has consistently achieved the best mean and median ranking among all surrogate-based methods in our experiments. These two methods are chosen because they represent both the established and state-of-the-art performance of surrogate-based approaches in offline optimization. As shown, although DEMO requires slightly more runtime than other selected methods, it achieves significantly higher effectiveness within the same time scale, as indicated by its superior mean ranking. It is noteworthy that Tri-mentoring exhibits similar time efficiency to DEMO, which may be attributed to its use of pairwise ranking information and bi-level optimization. These results demonstrate that DEMO not only mitigates the out-of-distribution challenges in offline model-based optimization but does so with practical computational demands, making it a viable solution for real-world applications.

## 6    Conclusion and Discussion

In this study, we introduce *$D$esign $E$diting for Offline $M$odel-based $O$ptimization* (**DEMO**). DEMO begins by training a surrogate model on the offline dataset as an estimate of the ground-truth black-box objective function. A batch of pseudo design candidates is then generated by performing gradient ascent with respect to the surrogate model. Subsequently, a diffusion model is trained on the offline dataset to capture the distribution of all existing valid designs. The design editing process introduces random noise to the pseudo design candidates and employs the learned diffusion model to denoise them, ensuring that the final optimized designs are not far from the prior distribution and have valid high scores. In essence, DEMO generates new designs by first extensively optimizing pseudo design candidates and then refining them with the diffusion prior. Experiments on the design-bench dataset show that, with properly tuned hyperparamters, DEMO's score is competitive with the best previously reported scores in the literature. We further discuss the connection between our method and Bayesian inference in Appendix A.6. One limitation of our proposed DEMO method is that its performance depends on the tuning of the hyperparameter $m$, which controls the amount of noise added during the design editing process, ensuring that the final design candidates are not extremely influenced by the extrapolation error of the surrogate while also avoiding an overly conservative edit that would revert them back to the low-scoring regime. It is important to note that such hyperparameter tuning for regularizations to balance overestimation and overconservatism is not unique to our approach; other methods in offline MBO may face similar challenges. Nonetheless, our experiments show that when $m$ is appropriately tuned, DEMO achieves a favorable trade-off and delivers competitive performance. We discuss additional limitations and potential negative impacts in Appendix A.7 and Appendix A.8, respectively.

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

Table 3: Experimental results on continuous tasks for comparison.

| Method | Superconductor | Ant Morphology | D'Kitty Morphology | Levy |
|---|---|---|---|---|
| $\mathcal{D}(\textbf{best})$ | 0.399 | 0.565 | 0.884 | 0.613 |
| BO-qEI | $0.300 \pm 0.015$ | $0.567 \pm 0.000$ | $0.883 \pm 0.000$ | $0.643 \pm 0.009$ |
| CMA-ES | $0.379 \pm 0.003$ | $-0.045 \pm 0.004$ | $0.684 \pm 0.016$ | $0.410 \pm 0.009$ |
| REINFORCE | $\textbf{0.463} \pm \textbf{0.016}$ | $0.138 \pm 0.032$ | $0.356 \pm 0.131$ | $0.377 \pm 0.065$ |
| Mean | $0.334 \pm 0.004$ | $0.569 \pm 0.011$ | $0.876 \pm 0.005$ | $0.561 \pm 0.007$ |
| Min | $0.364 \pm 0.030$ | $0.569 \pm 0.021$ | $0.873 \pm 0.009$ | $0.537 \pm 0.006$ |
| COMs | $0.316 \pm 0.024$ | $0.564 \pm 0.002$ | $0.881 \pm 0.002$ | $0.511 \pm 0.012$ |
| ROMA | $0.370 \pm 0.019$ | $0.477 \pm 0.038$ | $0.854 \pm 0.007$ | $0.558 \pm 0.003$ |
| NEMO | $0.320 \pm 0.008$ | $0.592 \pm 0.000$ | $0.883 \pm 0.000$ | $0.538 \pm 0.006$ |
| BDI | $0.412 \pm 0.000$ | $0.474 \pm 0.000$ | $0.855 \pm 0.000$ | $0.534 \pm 0.003$ |
| IOM | $0.350 \pm 0.023$ | $0.513 \pm 0.035$ | $0.876 \pm 0.006$ | $0.562 \pm 0.007$ |
| ICT | $0.399 \pm 0.012$ | $0.592 \pm 0.025$ | $0.874 \pm 0.005$ | $0.691 \pm 0.009$ |
| Tri-mentoring | $0.355 \pm 0.003$ | $0.606 \pm 0.007$ | $0.866 \pm 0.001$ | $0.687 \pm 0.012$ |
| PGS | $0.379 \pm 0.016$ | $0.532 \pm 0.016$ | $\textbf{0.941} \pm \textbf{0.008}$ | $0.476 \pm 0.014$ |
| CbAS | $0.111 \pm 0.017$ | $0.384 \pm 0.016$ | $0.753 \pm 0.008$ | $0.479 \pm 0.020$ |
| Auto CbAS | $0.131 \pm 0.010$ | $0.364 \pm 0.014$ | $0.736 \pm 0.025$ | $0.499 \pm 0.022$ |
| MIN | $0.336 \pm 0.016$ | $0.618 \pm 0.040$ | $0.887 \pm 0.004$ | $0.681 \pm 0.030$ |
| DDOM | $0.346 \pm 0.009$ | $0.615 \pm 0.007$ | $0.861 \pm 0.003$ | $0.595 \pm 0.012$ |
| BONET | $0.369 \pm 0.015$ | $\textbf{0.819} \pm \textbf{0.032}$ | $0.907 \pm 0.020$ | $0.604 \pm 0.008$ |
| $\textbf{DEMO}_{(\text{ours})}$ | $0.400 \pm 0.007$ | $0.604 \pm 0.005$ | $\textbf{0.891} \pm \textbf{0.002}$ | $\textbf{0.762} \pm \textbf{0.008}$ |

Table 4: Experimental results on discrete tasks, and ranking on all tasks for comparison.

| Method | TF Bind 8 | TF Bind 10 | NAS | Rank Mean | Rank Median |
|---|---|---|---|---|---|
| $\mathcal{D}(\textbf{best})$ | 0.439 | 0.467 | 0.436 | | |
| BO-qEI | $0.439 \pm 0.000$ | $0.467 \pm 0.000$ | $0.544 \pm 0.099$ | 9.4/19 | 10/19 |
| CMA-ES | $0.537 \pm 0.014$ | $0.484 \pm 0.014$ | $\textbf{0.591} \pm \textbf{0.102}$ | 10.6/19 | 7/19 |
| REINFORCE | $0.462 \pm 0.021$ | $0.475 \pm 0.008$ | $-1.895 \pm 0.000$ | 13.7/19 | 18/19 |
| Mean | $0.539 \pm 0.030$ | $\textbf{0.539} \pm \textbf{0.010}$ | $0.494 \pm 0.077$ | 8.3/19 | 8/19 |
| Min | $0.569 \pm 0.050$ | $0.485 \pm 0.021$ | $\textbf{0.567} \pm \textbf{0.006}$ | 7.3/19 | 8/19 |
| COMs | $0.439 \pm 0.000$ | $0.467 \pm 0.002$ | $0.525 \pm 0.003$ | 11.1/19 | 11/19 |
| ROMA | $0.555 \pm 0.020$ | $0.512 \pm 0.020$ | $0.525 \pm 0.003$ | 8.6/19 | 7/19 |
| NEMO | $0.438 \pm 0.001$ | $0.454 \pm 0.001$ | $\textbf{0.564} \pm \textbf{0.016}$ | 10.4/19 | 11/19 |
| BDI | $0.439 \pm 0.000$ | $0.476 \pm 0.000$ | $0.517 \pm 0.000$ | 10.4/19 | 10/19 |
| IOM | $0.439 \pm 0.000$ | $0.477 \pm 0.010$ | $-0.050 \pm 0.011$ | 11.0/19 | 10/19 |
| ICT | $0.551 \pm 0.013$ | $\textbf{0.541} \pm \textbf{0.004}$ | $0.494 \pm 0.013$ | 5.6/19 | $\textbf{4/19}$ |
| Tri-mentoring | $\textbf{0.609} \pm \textbf{0.021}$ | $0.527 \pm 0.008$ | $0.516 \pm 0.028$ | 6.1/19 | $\textbf{4/19}$ |
| PGS | $0.375 \pm 0.014$ | $0.443 \pm 0.005$ | $0.508 \pm 0.017$ | 12.0/19 | 12/19 |
| CbAS | $0.428 \pm 0.010$ | $0.463 \pm 0.007$ | $0.292 \pm 0.027$ | 16.3/19 | 16/19 |
| Auto CbAS | $0.419 \pm 0.007$ | $0.461 \pm 0.007$ | $0.217 \pm 0.005$ | 16.9/19 | 17/19 |
| MIN | $0.421 \pm 0.015$ | $0.468 \pm 0.006$ | $0.433 \pm 0.000$ | 9.3/19 | 12/19 |
| DDOM | $0.401 \pm 0.008$ | $0.464 \pm 0.006$ | $0.306 \pm 0.017$ | 11.9/19 | 13/19 |
| BONET | $0.505 \pm 0.055$ | $0.496 \pm 0.037$ | $0.571 \pm 0.095$ | 4.6/19 | 5/19 |
| $\textbf{DEMO}_{(\text{ours})}$ | $0.533 \pm 0.010$ | $0.480 \pm 0.003$ | $\textbf{0.564} \pm \textbf{0.005}$ | $\textbf{4.4/19}$ | $\textbf{4/19}$ |

# A  Appendix

## A.1  Median Normalized Scores

**Performance in Continuous Tasks.** Table 3 showcases the median normalized scores for various baseline methods across 4 continuous tasks. DEMO, while not always topping the charts, demonstrates robust performance across these tasks, consistently outperforming several baseline methods. For example, in the Levy function task, DEMO's score of $0.762 \pm 0.008$ is the highest one among all approaches. This highlights DEMO's capability to mitigate the OOD issue of surrogates based methods effectively. Notably, DEMO

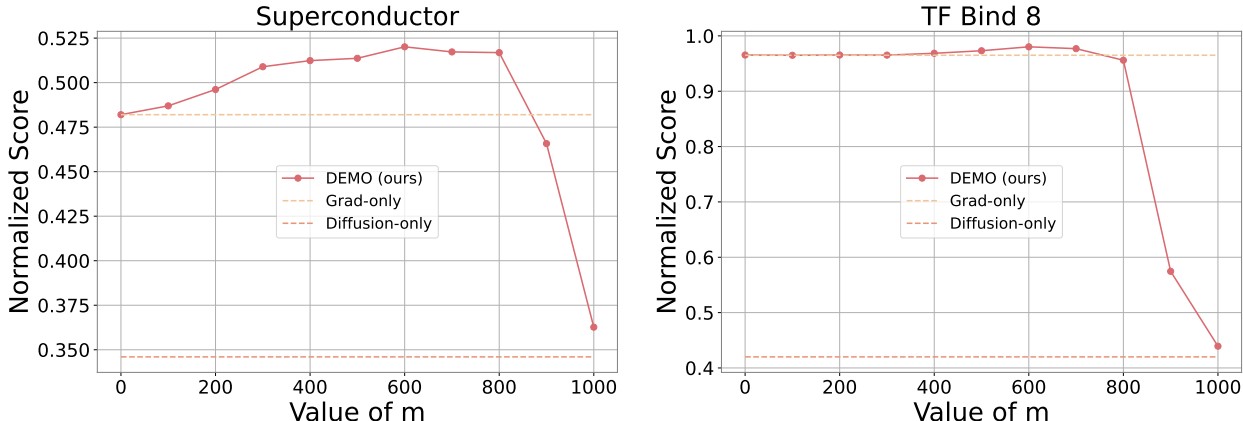

Figure 4: Selecting $m$ near 0 results in generated designs that retains most properties of pseudo design candidates. Conversely, setting $m$ near 1000 generates designs that align closely with the distribution of existing designs. Optimal designs are achieved by choosing $m$ in the mid-range, effectively utilizing information from both the pseudo design candidates and the diffusion prior.

outperforms traditional generative models like CbAS and Auto CbAS by significant margins across all tasks. It also maintains a competitive edge against more recent generative methods like MIN and DDOM.

**Performance in Discrete Tasks.** Moving to discrete tasks, as detailed in Table 4, DEMO exhibits performance on par with other baseline methods. This performance can be attributed to DEMO's methodology which, although highly effective in calibrating the pseudo design candidates, might struggle in task environments with redundancy in design features.

**Summary.** The results presented in Tables 3 and 4 collectively validate DEMO's efficacy across both continuous and discrete optimization tasks, providing further support for answering $Q1$ affirmatively. With a mean rank of 4.4/19 and a median rank of 4/19 in terms of the median normalized scores, DEMO stands out among 19 competing methods. This comprehensive performance underscores DEMO's capacity to integrate and leverage information from the distribution of existing designs and pseudo design candidates.

## A.2 Sensitivity to the Choice of m

In Eq. (9), selecting a time $m$ close to $M$ results in $\boldsymbol{x}_{perturb}$ resembling random Gaussian noise, which introduces greater flexibility into the new design generation process. On the other hand, if $m$ is closer to 0, the resulting design retains more characteristics of the pseudo design candidates. Thus, $m$ serves as a critical hyperparameter in our methodology. This section explores the robustness of DEMO to various choices of $m$. We perform experiments on one continuous task, SuperC, and one discrete task, TF8, with $m$ ranging from 0 to 1000 in increments of 100. As illustrated in Figure 4, DEMO generally outperforms the Diffusion-only and Grad-only methods. Nevertheless, overly extreme values of $m$, whether too high or too low, can diminish performance. Selecting an excessively low $m$ causes the model to adhere too closely to the pseudo design candidates, while choosing an overly high $m$ biases the model towards the distribution of existing designs, neglecting the guidance of pseudo design candidates. Choosing $m$ from a mid-range effectively balances the influences from both the prior distribution and the pseudo design candidates, leading us to set $m = 600$ for all tasks.

## A.3 Details of Corrected P-Values

As mentioned in the main text, all reported p-values are adjusted using the Bonferroni correction to account for multiple hypothesis testing. Specifically, each uncorrected p-value is multiplied by the total number of null hypotheses, which corresponds to the number of baseline methods (18). Table 5 presents the corrected p-values from comparisons between DEMO and other baseline methods. We bold the values where DEMO demonstrates statistically significant improvement over the corresponding baseline method at a significance

Table 5: Corrected p-values on all tasks.

| Method | Superconductor | Ant | D'Kitty | Levy | TF8 | TF10 | NAS |
|---|---|---|---|---|---|---|---|
| BO-qEI | **0.000** | **0.000** | **0.000** | **0.000** | **0.003** | **0.007** | 0.000 |
| CMA-ES | **0.001** | 3.362 | **0.000** | **0.000** | 0.091 | **0.021** | 0.000 |
| REINFORCE | **0.000** | **0.000** | **0.005** | **0.000** | 0.111 | **0.013** | **0.000** |
| Mean | **0.032** | **0.004** | 0.264 | 0.319 | **0.000** | **0.007** | **0.025** |
| Min | 0.081 | **0.035** | **0.000** | **0.007** | **0.042** | **0.002** | **0.017** |
| COMs | **0.023** | **0.000** | **0.002** | **0.000** | **0.000** | **0.002** | 0.035 |
| ROMA | 0.221 | **0.013** | **0.000** | **0.027** | **0.027** | **0.021** | 0.000 |
| NEMO | **0.001** | **0.046** | **0.000** | **0.006** | **0.001** | 0.306 | 0.269 |
| BDI | 0.063 | **0.000** | **0.000** | **0.000** | **0.000** | **0.001** | **0.012** |
| IOM | 3.496 | **0.027** | **0.002** | 0.524 | **0.030** | **0.005** | **0.000** |
| ICT | 0.075 | 0.952 | 7.162 | **0.000** | **0.030** | 0.093 | 0.289 |
| Tri-mentoring | 1.370 | **0.048** | 3.344 | **0.001** | 0.645 | 0.875 | 7.866 |
| PGS | 0.970 | 0.162 | 4.140 | **0.018** | 8.093 | **0.010** | 0.670 |
| CbAs | 3.600 | **0.000** | **0.000** | **0.007** | 0.168 | **0.019** | 0.371 |
| Auto CbAs | **0.002** | **0.008** | **0.000** | **0.000** | **0.017** | **0.002** | **0.000** |
| MIN | **0.027** | **0.000** | **0.000** | **0.000** | **0.033** | **0.001** | 0.613 |
| DDOM | **0.000** | **0.014** | **0.000** | **0.001** | 0.553 | **0.003** | 0.538 |
| BONET | **0.000** | 1.398 | 0.067 | **0.000** | 2.383 | 0.052 | **0.013** |
| **DEMO**(ours) | 10 | 14 | 13 | 16 | 11 | 14 | 7 |

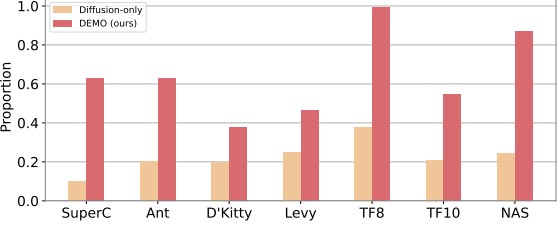

Figure 5: The proportion is calculated as the number of new designs which surpass $\mathcal{D}(\mathbf{best})$ divided by the budget 128, indicating the reliability to consistently generate new higher-scoring designs. This figure demonstrates that DEMO is more reliable than Diffusion-only in all tasks.

Figure 6: The proportion is calculated as the number of new designs which surpass $\mathcal{D}(\mathbf{best})$ divided by the budget 128, indicating the reliability to consistently generate new higher-scoring designs. This figure demonstrates that DEMO is more reliable than Grad-only in 5/7 tasks.

level of $\alpha = 0.05$. The last row summarizes the number of baseline methods for which DEMO achieves statistically significant improvement in each task.

## A.4 Reliability Study

In this subsection, we assess the ability of DEMO to reliably produce superior designs compared to selected surrogate based methods and generative model based methods. To measure reliability, we compute the proportion of new designs that exceed the best scores in the offline dataset $\mathcal{D}(\mathbf{best})$. The results are depicted in Figure 5. DEMO consistently outperforms Diffusion-only across all tasks, achieving notable improvements, particularly in the SuperC and NAS tasks. This confirms DEMO's enhanced reliability over the state-of-the-art generative model-based baseline in both continuous and discrete settings. We then extends the reliability study to compare DEMO with a gradient-based approach. When compared to Grad-only, DEMO demonstrates greater consistency in 5 out of 7 tasks. However, Grad-only outperforms DEMO in Levy and TF10 tasks, which can be attributed to the gradient-based method's tendency to generate new designs within a narrower distribution. While Grad-only achieves a higher proportion of higher-scoring new designs in these two tasks, DEMO generates new designs within a wider distribution and thus produces candidates with higher maximum scores, as evidenced in Table 2.

## A.5 Quantitative Analysis and Convergence Study

Table 6: Quantitative Analysis and Convergence Study for Ant

| Gradient Ascent Step | 100 | 200 | 300 |
|---|---|---|---|
| Predicted Score of Pseudo Design Candidates | 0.950 | 1.210 | 1.340 |
| Ground-truth Score of Pseudo Design Candidates | 0.942 | 0.887 | 0.845 |
| Ground-truth Score of Edited Final Design Candidates | 0.968 | 0.935 | 0.928 |

Table 7: Quantitative Analysis and Convergence Study for TF8

| Gradient Ascent Step | 100 | 200 | 300 |
|---|---|---|---|
| Predicted Score of Pseudo Design Candidates | 2.916 | 5.570 | 8.311 |
| Ground-truth Score of Pseudo Design Candidates | 0.912 | 0.895 | 0.895 |
| Ground-truth Score of Edited Final Design Candidates | 0.982 | 0.958 | 0.914 |

We run additional experiments on one continuous task Ant and one discrete task TF8. As shown in Table 6 and Table 7, the pseudo design candidates usually have over optimistic predicted score, but their ground-truth scores evaluated by the oracle are not such high. After editing the pseudo design candidates by our approach, the ground-truth scores are effectively increased.

In our original setting, we run 100 gradient ascent steps to acquire the pseudo design candidates. 200 and 300 gradient ascent steps are considered in the additional experiments, where we can mimic the distribution very far away from the training distribution. As we can see from Table 6 and Table 7, even though the performance degrades as the number of gradient ascent steps increases, our proposed approach is still helpful for editing the pseudo design candidates and thus improving the quality of the final candidates.

## A.6 Connection to Bayesian Inference

From a Bayesian perspective, our objective is to find the design $\boldsymbol{x}$ that maximizes the posterior probability:

$$p(\boldsymbol{x}|y) \propto p(y|\boldsymbol{x})p(\boldsymbol{x}), \tag{10}$$

where $p(y|\boldsymbol{x})$ is the likelihood of obtaining a property score $y$ given a design $\boldsymbol{x}$, and $p(\boldsymbol{x})$ is the prior distribution over designs learned from the offline dataset. Taking the logarithm, we have:

$$\log p(\boldsymbol{x}|y) = \log p(y|\boldsymbol{x}) + \log p(\boldsymbol{x}) + C, \tag{11}$$

with $C$ being a constant that does not affect the optimization. To find the design that maximizes the posterior, we aim to solve:

$$\boldsymbol{x}^* = \arg\max_{\boldsymbol{x} \in \mathcal{X}}[\log p(y|\boldsymbol{x}) + \log p(\boldsymbol{x})]. \tag{12}$$

In the offline model-based optimization (MBO), we typically do not have a direct expression for $p(y|\boldsymbol{x})$. Instead, we train a surrogate model $f_{\boldsymbol{\theta}}(\boldsymbol{x})$ to predict the property score $y$. By following Lee et al. (2023); Yuan et al. (2024), we can model the probability density using the Boltzmann distribution as follows:

$$p(y|\boldsymbol{x}) \approx p_{\boldsymbol{\theta}}(y|\boldsymbol{x}) = \frac{e^{\gamma f_{\boldsymbol{\theta}}(\boldsymbol{x})}}{Z}, \tag{13}$$

where $\gamma$ is the scaling factor, and $Z$ is the normalization constant. Taking the logarithm of this expression, we obtain:

$$\log p(y|\boldsymbol{x}) \approx \gamma f_{\boldsymbol{\theta}}(\boldsymbol{x}) - \log Z. \tag{14}$$

To maximize the posterior, we wish to solve Eq. 12. In practice, we can perform gradient ascent. By differentiating the log-posterior, the update rule becomes:

$$\boldsymbol{x}_{t+1} = \boldsymbol{x}_t + \eta \, \nabla_{\boldsymbol{x}} \left[ \log p(y|\boldsymbol{x}) + \log p(\boldsymbol{x}) \right] \Big|_{\boldsymbol{x}=\boldsymbol{x}_t}, \tag{15}$$

where $\eta$ is the learning rate. Since $p(y|\boldsymbol{x})$ is approximated by the surrogate model as discussed above, we have:

$$\nabla_{\boldsymbol{x}} \log p(y|\boldsymbol{x}) \approx \nabla_{\boldsymbol{x}} f_{\boldsymbol{\theta}}(\boldsymbol{x}), \tag{16}$$

and the diffusion model is trained to capture the prior, yielding an approximation:

$$\nabla_{\boldsymbol{x}} \log p(\boldsymbol{x}) \approx s_{\boldsymbol{\phi}}(\boldsymbol{x}, 0), \tag{17}$$

where $s_{\boldsymbol{\phi}}(\boldsymbol{x}, 0)$ is the score function learned by the diffusion model on the original data at time 0. At $t = 0$, no noise has been added, so the score function ideally estimates the gradient of the log probability of the original data distribution. It is worth noting that, in practice, diffusion models are trained to learn the score function over a continuous range of time steps. Although $t = 0$ represents the ideal scenario, a small nonzero $t$ should be used for numerical stability. This update can be interpreted as performing approximate maximum a posteriori (MAP) estimation, where the first term drives the design toward higher predicted scores, namely maximizing the likelihood. The second term enforces that the design remains within the valid design manifold, that is incorporating the prior.

In Bayesian perspective, DEMO can be seen as performing an approximate posterior inference in two steps: (1) Posterior Approximation, that is the acquirement of pseudo design candidates, using the surrogate to imagine where high-score probability mass might lie, and (2) Posterior Refinement, namely the design editing process, using the diffusion prior to correct and refine those candidates. As detailed above, while a single-phase Bayesian inference formulation offers an elegant theoretical framework by directly optimizing the combined log-posterior, our two-phase approach in DEMO provides practical advantages. In the single-phase formulation, the update rule simultaneously balances the gradient of the likelihood, derived from the surrogate model, and the gradient of the prior, obtained via the diffusion model, within a single unified update. This balance can be challenging to achieve in practice, which may require additional hyperparameters to trade-off. More importantly, directly applying the diffusion prior in the update rule may not be enough to edit the out-of-distribution candidates back to the valid range. In contrast, our two-phase method decouples these tasks: the first phase focuses exclusively on generating high-scoring candidates via gradient ascent on the surrogate model, while the second phase leverages the diffusion model to edit and refine these candidates, ensuring they remain within the valid design manifold. This modular design simplifies the optimization process by isolating each objective, achieving a more stable and effective balance between high performance and design realism compared to a direct single-phase Bayesian optimization.

## A.7 Limitations

We have demonstrated the effectiveness of DEMO across a wide range of tasks. However, some evaluation methods may not fully capture real-world complexities. For example, in the superconductor task (Hamidieh, 2018), we follow traditional practice by using a random forest regression model as the oracle, as done in prior studies (Trabucco et al., 2022). Unfortunately, this model might not entirely reflect the intricacies of real-world situations, which could lead to discrepancies between our oracle and actual ground-truth outcomes. Engaging with domain experts in the future could help enhance these evaluation approaches. Nevertheless, given DEMO's straightforward approach and the empirical evidence supporting its robustness and efficacy across various tasks detailed in the Design-Bench (Trabucco et al., 2022) and BayesO Benchmarks (Kim, 2023), we remain confident in its ability to generalize effectively to different contexts.

## A.8 Negative Impacts

This study seeks to advance the field of Machine Learning. However, it's important to recognize that advanced optimization techniques can be used for either beneficial or detrimental purposes, depending on their application. For example, while these methods can contribute positively to society through the development

of drugs and materials, they also have the potential to be misused to create harmful substances or products. As researchers, we must stay aware and ensure that our contributions promote societal betterment, while also carefully assessing potential risks and ethical concerns.

