# OpenReview forum: "Design Editing for Offline Model-based Optimization"
_TMLR — Accepted by TMLR_

### Review · Reviewer_5gy3 · 2025-02-07

**Summary Of Contributions:**

This paper proposes Design Editing for Offline Model-based Optimization (DEMO). It is essentially a two stage procedure for offline black-box optimization:

1. A surrogate model $\hat f$ is trained on offline data, and then a candidate point $\tilde x^*\in\arg\max_x \hat f(x)$ is chosen
2. A diffusion model is learned to diffuse $\tilde x^*$ closer to the data manifold, producing a final candidate $x^*$

The key assumption is that points closer to the data manifold are less likely to be "spurious optima" of the surrogate model (note: I think the paper would be improved if the authors explained this intuition more clearly). The authors evaluate their method on design-bench and find that it performs competitively.

**Audience:**

Yes

**Broader Impact Concerns:**

No concerns

**Claims And Evidence:**

No

**Requested Changes:**

I think the empirical evidence does not support the claim from the introduction that:

> Extensive experiments demonstrate DEMO effectively and reliably generates new designs, yielding state-of-the-art results across 7 offline MBO tasks, with the mean rank of 2.1 and the median rank of 1 among 19 methods.

In my opinion, a more honest summary of the results would be:

> Experiments on the design-bench dataset show that, with properly tuned hyperparamters, DEMO's score is competitive with the best previously reported scores in the literature

To explain the differences:

- I think it is an exaggeration to call design bench "extensive experiments"
- The "state-of-the-art results" claims seem exaggerated, hence "competitive" instead of "state-of-the-art":
   - Many performance differences seem small and likely not statistically significant (the most egregious example I spotted was TF Bind-8, where #1 rank is claimed for 0.982±0.016, compared with 0.981±0.015 for PGS).
  - The t-test results seem wrong. For example, a p-value of 0.03 is claimed on TF8, despite the difference in means only being $10^{-3}$ and the standard errors of each estimate being $\approx 10^{-2}$. It also looks like no correction for multiple tested hypotheses was done.
- The results are obtained for only one set of hyperparameters (e.g. training length, weight decay coefficients, number of gradient steps to obtain candidate point, diffusion schedule). From my experience in ML I guess that the results are quite sensitive to the settings of these parameters.

If the authors would like to maintain their original claims I think more experimental evidence would be necessary.

Finally, please write $10^{-3}$ instead of $1e-3$:

$1e-3=e-3\approx -0.281718$.

**Strengths And Weaknesses:**

Strengths:

- Idea in the paper is reasonable
- Experimental results are promising (although I have some issues with the precise claims)
- Paper is reasonably well-written

Weaknesses:

- I don't think the empirical results fully support the claims about DEMO being state-of-the-art, first because the results don't all seem statistically significant, and second because the method is only tested at a single setting of hyperparameters. More details in "requested changes"
- Offline MBO is a field full of very similar methods, which all try to generate high-scoring points, but regularized to be near the data manifold. A deeper analysis and explanation of DEMO's approach vs the approach of baseline methods would be nice. At the moment, §3 basically just mentions the existence of many papers.
- This is somewhat of a personal opinion, but
    1. I don't think offline MBO is a realistic problem (specifically, in any of the applications mention, is there really only one opportunity to generate designs, because if it is ≥2 then it is not offline anymore)
    2. I don't think design bench tasks are particularly realistic (eg, in all my years of machine learning I have never heard of anybody doing offline neural architecture search).

---

> ### Author Response · Authors · 2025-02-19
> **Reply to Reviewer 5gy3**
>
> ## Genernal Reply
> We would like to thank Reviewer 5gy3 for the thorough evaluation of our manuscript and for highlighting both its strengths and areas for improvement.
> We appreciate the time and effort invested in reviewing our work.
> We also apologize for the delay of our responses.
> Below, we address each of the reviewer’s comments in detail and outline how we will incorporate the suggested changes and clarifications.
>
> ## Concerns from Reviewer 5gy3
> > note: I think the paper would be improved if the authors explained this intuition more clearly.
>
> We thank for the constructive suggestion from the reviewer.
> To explicitly and more clearly explain the intuition behind our method, we would like to add the following paragraph to the introduction section, which is highlighted in blue in the revised manuscript:
>
> "A central assumption underlying our approach is that candidate designs located near the offline data manifold are less prone to being “spurious optima” of the surrogate model.
> In regions far from the observed data, the model must extrapolate, which often leads to unreliable predictions and artificially inflated performance estimates.
> Essentially, the model may identify a high-scoring design that, in reality, is merely an artifact of overfitting or model bias.
> By contrast, when candidate points remain close to the data manifold, they are supported by the training data and the model’s predictions in these regions are more trustworthy.
> Therefore, by guiding our search toward these well-supported regions, we reduce the risk of selecting designs that appear optimal only due to the model’s extrapolative errors, and instead focus on candidates that are both high-performing and realistically feasible."
>
> > I don't think the empirical results fully support the claims about DEMO being state-of-the-art, first because the results don't all seem statistically significant, and second because the method is only tested at a single setting of hyperparameters. More details in "requested changes"
>
> We acknowledge that the more honest summary of experiment results suggested by the reviewer is more proper than the original claim in our paper.
> We expand the discussion in the responses to requested changes below.
>
> > Offline MBO is a field full of very similar methods, which all try to generate high-scoring points, but regularized to be near the data manifold. A deeper analysis and explanation of DEMO's approach vs the approach of baseline methods would be nice. At the moment, §3 basically just mentions the existence of many papers.
>
> We acknowledge that a deeper analysis and explanation of DEMO's approach versus existing baseline methods would be better.
> Therefore, we make the following modifications to section 3.1, which are hilighted in blue in the revised manuscript.
> (1) We first summarize the gradient-based approaches in this way: "Gradient-based methods directly optimize the surrogate model's predictions using gradient-based search. However, straightforward gradient ascent can lead to candidate designs that are far from the observed data, where the surrogate is prone to unreliable extrapolations."
> (2) And we add the following paragraph to the end of section 3.1 in addition to just mentioning existing approaches: "In DEMO, a candidate design is first generated via gradient-based optimization, ensuring that the candidate is high-scoring according to the surrogate model.
> Recognizing that direct gradient-based updates can push the candidate into regions where the model’s predictions are unreliable, we then employ a diffusion model to “edit” the design, explicitly moving it closer to the data manifold.
> This two-stage process decouples the search for high performance from the regularization required to maintain realism.
> While traditional gradient-based methods embed regularization within the optimization loop and generative methods rely solely on learned distributions, DEMO leverages the flexibility of gradient-based search alongside the robustness of diffusion-based editing.
> This approach effectively reduces the risk of spurious optima-candidates that appear optimal due to extrapolative errors, by ensuring that final designs remain in regions well-supported by data.
> In summary, while existing offline MBO methods typically focus on either direct gradient-based optimization with embedded regularization or on generative modeling of the design space, DEMO’s hybrid approach offers a novel balance between performance and realism, thereby addressing key challenges inherent in both paradigms."

---

> ### Author Response · Authors · 2025-02-19
> **Additional Reply to Reviewer 5gy3**
>
> > This is somewhat of a personal opinion, but (1) I don't think offline MBO is a realistic problem (specifically, in any of the applications mention, is there really only one opportunity to generate designs, because if it is greater than or equal to 2 then it is not offline anymore) (2) I don't think design bench tasks are particularly realistic (eg, in all my years of machine learning I have never heard of anybody doing offline neural architecture search).
>
> We thank the reviewer for their candid opinions. We respectfully offer the following clarifications:
> (1) Realism of Offline MBO: Numerous works in domains such as drug discovery, materials design, and industrial process optimization have demonstrated the challenges and practical importance of optimizing designs when further queries to an oracle are prohibitively expensive or simply unavailable.
> Even in settings where multiple rounds of queries are possible, the final round remains offline since no additional oracle feedback is available after the ultimate design is selected.
> This scenario is particularly relevant in high-stakes applications where experimental evaluations are costly or time-consuming.
> Moreover, as discussed in the related work section of our manuscript, offline MBO is indeed a recognized problem and active research area within the machine learning and optimization communities.
> (2) Realism of Design Bench Tasks: While we acknowledge that the specific tasks in Design-Bench may not perfectly mirror every aspect of real-world design challenges (such as offline neural architecture search), we still benchmark our method on them for two main reasons. (i) they have emerged as a standardized benchmark within the offline MBO community. The use of these benchmarks enables consistent, reproducible comparisons across various methods. Many contemporary works in the field evaluate their methods on these tasks, which facilitates a fair comparison and accelerates progress in addressing the inherent challenges of offline MBO. (ii) Even though some tasks are traditionally addressed via an online approach, converting them into an offline setting offers significant advantages when querying the oracle is expensive or time-consuming. This not only saves computational resources but also accelerates the optimization process.
>
>
>
> ## Requested Changes
> > I think the empirical evidence does not support the claim from the introduction that...
>
> As stated earlier, we acknowledge that the more honest summary of experiment results suggested by the reviewer is more proper than the original claim in our paper.
> Therefore, we update our claim in the revised version and highlight it in blue by stating "Experiments on the design-bench dataset show that, with properly tuned hyperparamters, DEMO's score is competitive with the best previously reported scores in the literature."
> Meanwhile, all related claims are all updated in the revised version, which are highlighted in blue as well.
>
> Regarding the t-test results, the p-values are obtained by comparing the results of DEMO and the best results of unbolded methods, since all bolded methods are considered as competitive to each other.
> To clarify this, we further explain in section 5.5 by stating that "We also conduct a Welch's t-test on the tasks where DEMO achieved competitive results, where DEMO is compared with the best unbolded results."
>
> For the settings of hyperparameters, we follow existing works to set the training length, weight decay coefficients etc. for a fair comparison.
> Nevertheless, we acknowledge that some specific hyperparameters, like m, are important to our approach, as it controls the trade-off between performance and realism.
>
> Consequently, we agree the suggested modifications by the reviewer are more proper than our original claims, and thus make the changes accordingly.
>
> > Finally, please write $10^{-3}$ instead of $1e-3$
>
> Thanks for pointing out this issue. We have already corrected them in the revised version and highlighted in blue.

---

> ### Author Response · Authors · 2025-03-04
> **Looking Forward to Your Feedback**
>
> Dear Reviewer 5gy3,
>
> We sincerely appreciate your detailed and thoughtful feedback, which has been instrumental in refining our manuscript. Your comments have helped us improve the clarity of our intuition, the depth of our comparative analysis, and the transparency of our experimental results.
>
> To address your concerns, we have carefully incorporated the following updates into our revised manuscript:
>
> - We provide a clearer explanation of the intuition behind our method in the introduction, emphasizing the role of diffusion-based editing in mitigating extrapolation errors (highlighted in blue).
> - We refine our empirical claims to ensure a more honest and accurate summary of our results, making clear that DEMO achieves competitive performance with properly tuned hyperparameters.
> - We expand Section 3.1 with a deeper comparison between DEMO and existing offline MBO methods, highlighting its unique two-phase approach.
> - We clarify our statistical significance tests and explicitly describe how Welch’s t-test is applied in Section 5.5.
> - We revise the related work section to better position DEMO within the broader field. We discuss the concerns about the realism of offline MBO and design-bench tasks in our previous replies.
> - We have corrected all formatting issues you pointed out.
>
> We greatly appreciate your time and insights. We wanted to check if there are any additional concerns or clarifications needed from our side. We look forward to your response and further discussion.
>
> Best regards,
>
> Authors

---

> > ### Comment · Reviewer_5gy3 · 2025-03-04
> > **Thanks for the update. Only concern is multiple hypothesis test.**
> >
> > Thanks for the updated paper. I have read your revised manuscript and the comments by the other reviewers.
> >
> > - I appreciate the revision of the empirical claims, thanks for this!
> > - I appreciate the typographical corrections
> > - I appreciate discussion of extra related work and the intuitive explanation
> > - I know another reviewer asked for a theoretical interpretation of the method. I _do not_ think this is necessary and that the reviewer maybe misunderstood the TMLR reviewing guidelines...
> >
> > My only remaining concern is the t-test results. You claim to only test against the "best" non-bolded method. However, you do not know the best non-bolded method, because you only have random observations and do not know the true mean. By comparing against the method with the highest _empirical_ mean (which I presume is what you meant), you are implicitly performing multiple hypothesis tests. I believe you should follow statistical best-practices here and perform a multiple correction for multiple hypothesis tests. The most straightforward such correction is the Bonferroni correction. Could you please adjust your results to use Bonferroni or some other appropriate multiple hypothesis test?

---

> > > ### Author Response · Authors · 2025-03-05
> > > **Reply to the concern of multiple hypotheses test**
> > >
> > > We thank the reviewer for the valuable suggestion regarding the multiple hypothesis testing issue, which we may misunderstand before. We have updated our manuscript accordingly and highlight them in blue. Specifically, we now state:
> > >
> > > "We also conduct a Welch’s t-test on the six tasks where DEMO achieved competitive results, where DEMO is compared with the unbolded methods with the highest empirical means. We obtain p-values of 0.021 on SuperC, 0.015 on Ant, 0.022 on D’Kitty, 0.009 on Levy, 0.184 on TF8, and 0.031 on TF10. Since we implicitly perform multiple hypothesis tests, all reported p-values are corrected with the Bonferroni correction to control the family-wise error rate. When setting the significance level α = 0.05, these hypotheses tests confirm that DEMO achieves statistically significant improvements in 5 out of 7 tasks."
> > >
> > > Could you please let us know if this revision addresses your concerns, or if there are any further aspects that we could improve more?
> > >
> > > Best regards,
> > >
> > > Authors

---

> > > > ### Comment · Reviewer_5gy3 · 2025-03-05
> > > > **Did you do this correctly?**
> > > >
> > > > Thank you for doing this, but I am worried that you did it incorrectly because your p-values went _down_, not _up_. Can you provide a sample calculation to show that this was done correctly?

---

> > > > > ### Author Response · Authors · 2025-03-05
> > > > > **Reply to the concern about the correct implementation of Bonferroni correction**
> > > > >
> > > > > Dear Reviewer,
> > > > >
> > > > > Thanks for pointing out this issue. We indeed mistakenly calculate it wrong. To clarify our Bonferroni correction procedure: as you pointed out since we don't know the ground-truth mean and only know the empirical mean of all methods, comparing with the unbold method with the highest empirical mean is implicitly comparing against all unbold methods. Therefore, the number of null hypotheses is equal to the number of unbold methods in each task. As suggested by Bonferroni correction, a null hypothesis should be rejected if the p-value is smaller than or equal to the significance level divided by the number of null hypotheses. We thus correct the p-values by multiplying them with the number of unbold methods in each task.
> > > > >
> > > > > Take the superconductor task as an example. When comparing our DEMO with BDI (an unbold method with the highest empirical mean), the uncorrected p-value we obtained from the Welch's t-test is 0.003485. We then correct it by multipying 0.003485 with 14 (the number of unbold methods in superconductor, except D(best)) and obtain 0.049 as the corrected p-value, which is smaller than 0.05. We then think our proposed DEMO is statistically significant on the superconductor task. The same logic is applied to all tasks.
> > > > >
> > > > >
> > > > > We revise our claim in the manuscript to the following:
> > > > >
> > > > > "We also conduct a Welch’s t-test on the six tasks where DEMO achieved competitive results, where DEMO is compared with the unbolded methods with the highest empirical means. Since we implicitly perform multiple hypotheses tests, all reported p-values are corrected with the Bonferroni correction to control the family-wise error rate. We obtain the corrected p-values of 0.049 on SuperC, 0.039 on Ant, 0.052 on D’Kitty, 0.024 on Levy, 0.461 on TF8, and 0.082 on TF10.  When setting the significance level α = 0.05, these hypotheses tests confirm that DEMO achieves statistically significant improvements in three tasks"
> > > > >
> > > > > We would appreciate it if you could let us know whether our understanding is correct and this revision addresses your concerns, or if there are any additional areas that require further improvement.
> > > > >
> > > > > Best regards,
> > > > >
> > > > > Authors

---

> > > > > > ### Comment · Reviewer_5gy3 · 2025-03-06
> > > > > > **Check again?**
> > > > > >
> > > > > > Thank you for the quick revision. What you wrote seems to be correct (essentially multiplying the p-value by 14), but I'm still not understanding your numbers.
> > > > > >
> > > > > > You write:
> > > > > >
> > > > > > > We obtain the corrected p-values of 0.049 on SuperC, 0.039 on Ant, 0.052 on D’Kitty, 0.024 on Levy, 0.461 on TF8, and 0.082 on TF10.
> > > > > >
> > > > > > However, in the version from Feb 27, you wrote:
> > > > > >
> > > > > > > We obtain p-values of 0.20 on SuperC, 0.04 on Ant, 0.01 on D’Kitty, 0.03 on Levy, 0.03 on TF8, and 0.005 on TF10.
> > > > > >
> > > > > > Unless I am misunderstanding something, shouldn't all your p-values essentially be multiplied by 14? Why did SuperC go down by ~4 and Ant stay the same? TF8 and TF10 look as I would expect.

---

> > > > > > > ### Author Response · Authors · 2025-03-06
> > > > > > > **Reply to the Significance Test Concern**
> > > > > > >
> > > > > > > Thank you again for the prompt response.
> > > > > > >
> > > > > > > To clarify the concerns brought by the numbers: when recalculating the p-values by involving the Bonferroni correction, we notice that some methods are mistakenly bolded/unbolded in the main tables. We therefore rectified them all (as you may see the difference from the initial manuscript and the latest revised one). Since the uncorrected p-values are calculated by comparing our DEMO against the unbolded method with the highest empirical mean, these rectifications of boldness of methods in the main table may cause the changes to the uncorrected p-values as well.
> > > > > > >
> > > > > > > Another point is that we are not multiplying all uncorrected p-values by 14. As we mentioned previously, we may multiplying all uncorrected p-values by the number of unbold methods in each task. For example, it is 14 for the superconductor task but 15 for the Ant Morphology task.
> > > > > > >
> > > > > > > Hope these clarifications explain why there might be some unexpected changes of the corrected p-values.
> > > > > > >
> > > > > > > Best regards,
> > > > > > >
> > > > > > > Authors

---

> > > > > > > > ### Comment · Reviewer_5gy3 · 2025-03-06
> > > > > > > > **What is bolded?**
> > > > > > > >
> > > > > > > > Thanks for the clarification, this makes sense, but raises another question: what determined which methods in the table are in bold? I searched for the word "bold" in the manuscript but did not find an explanation. Sorry if it is there and I missed it.
> > > > > > > >
> > > > > > > > Essentially what I want to ask is "why does it make sense to compare DEMO with the best unbolded method"?

---

> > > > > > > > > ### Author Response · Authors · 2025-03-06
> > > > > > > > > **Further Clarifications added to our manuscript**
> > > > > > > > >
> > > > > > > > > Dear Reviewer,
> > > > > > > > >
> > > > > > > > > We also incorporate the following clarifications at the begining of section5.5 in the revised manuscript:
> > > > > > > > >
> > > > > > > > > "Following existing studies, in Table 1 and Table 2 we mark a method in bold if its mean is at least as high as the highest mean minus one standard deviation of the corresponding method. Alternatively, a method is also bolded if its mean plus one standard deviation reaches or exceeds the method with the highest mean."
> > > > > > > > >
> > > > > > > > > We hope these clarifications may help readers better understand what methods are bolded in the main tables.
> > > > > > > > >
> > > > > > > > > Could you kindly let us know whether this revision resolves your concerns or if there are any additional areas that need further improvement?
> > > > > > > > >
> > > > > > > > > Best regards,
> > > > > > > > >
> > > > > > > > > Authors

---

> > > > > > > > > > ### Comment · Reviewer_5gy3 · 2025-03-07
> > > > > > > > > > **Perhaps just test against all baseline?**
> > > > > > > > > >
> > > > > > > > > > Thank you for the clarification in the manuscript. I don't have any issues with the bolding in your table, since I consider it an cosmetic choice. However, the choice to do a statistical test between DEMO and the highest unbolded method seems strange then: aren't you essentially testing whether there is a significant difference between DEMO and the set of methods which don't appear to be statistically different? Essentially you remove any baseline which might make the difference insignificant. Is this essentially justifying your choice of bolding?
> > > > > > > > > >
> > > > > > > > > > Would it instead make more sense to compare with _all_ methods? Ie test for best performance?

---

> > > > > > > > > > > ### Author Response · Authors · 2025-03-08
> > > > > > > > > > > **Reply to test against all baseliens**
> > > > > > > > > > >
> > > > > > > > > > > Dear Reviewer,
> > > > > > > > > > >
> > > > > > > > > > > Thank you for your valuable suggestions. We fully agree that testing against all baselines provides a more comprehensive evaluation. In response, we have added a new section in Appendix A.3, which includes all corrected p-values from comparisons between DEMO and all baseline methods across all tasks.
> > > > > > > > > > >
> > > > > > > > > > > Furthermore, since we are now conducting comparisons against all baselines, we believe it is important to clearly state how DEMO performs relative to all other methods. To reflect this, we have revised our manuscript and now explicitly state:
> > > > > > > > > > >
> > > > > > > > > > > "We perform Welch’s t-test across the seven tasks to compare DEMO against all baseline methods. To account for multiple hypothesis tests, we apply the Bonferroni correction to all reported p-values to control the family-wise error rate. The complete set of corrected p-values is provided in Appendix A.3. In summary, at a significance level of α = 0.05, the results confirm that DEMO achieves statistically significant improvements over 10 baseline methods in the Superconductor task, 14 in the Ant task, 13 in the D’Kitty task, 16 in the Levy task, 11 in the TF8 task, 14 in the TF10 task, and 7 in the NAS task."
> > > > > > > > > > >
> > > > > > > > > > > We hope this revision provides a more comprehensive comparison of DEMO against all baselines and addresses your concerns.
> > > > > > > > > > >
> > > > > > > > > > > Best regards,
> > > > > > > > > > >
> > > > > > > > > > > Authors

---

> > > > > > > > > > > > ### Comment · Reviewer_5gy3 · 2025-03-10
> > > > > > > > > > > > **Happy with statistical testing**
> > > > > > > > > > > >
> > > > > > > > > > > > Thank you for changing the statistical tests. I have no further concerns about the methodology here. I have no further requests for this paper :)

---

> > > > > > > > > > > > > ### Author Response · Authors · 2025-03-10
> > > > > > > > > > > > > **Great thanks for the review**
> > > > > > > > > > > > >
> > > > > > > > > > > > > Dear reviewer 5gy3,
> > > > > > > > > > > > >
> > > > > > > > > > > > > We appreciate the opportunity to address all of your concerns. Thank you for taking the time to review our response and for your suggestions to clarify some claims in our manuscript. We sincerely value your thoughtful feedback and constructive insights.
> > > > > > > > > > > > >
> > > > > > > > > > > > > Best regards,
> > > > > > > > > > > > >
> > > > > > > > > > > > > Authors

---

> ### Author Response · Authors · 2025-03-06
> **Reply to the concern of boldness**
>
> Thanks for the prompt response again!
>
> From the existing literature, one common way people mark the boldness of each method is (1) first select the method with the highest empirical mean. (2) then substract one standard deviation from the empirical mean of that methods (3) mark all methods with empirical means higher than that value in bold. For example, in task TF Bind 8, DEMO has the highest empirical mean "0.982", we substract one standard deviation "0.016" from this value to obtain "0.966", and then mark all methods with empirical means higher or equal to 0.966 in bold.
>
> Another case we mark a method in bold is that its empirical mean add one standard deviation can reach (greater than or equal to) the methods with the highest empirical mean. For example, the "Mean" method in D'Kitty task has an empirical mean of "0.956" and standard deviation of "0.014", we get "0.970" when added them up, which is equal to the empirical mean of DEMO (the highest method in this task). Similar logic is applied to methods like "CbAS" in superconductor and "Mean" in levy.
>
> Sometimes we make some exceptions when a method has very large standard deviation. For example, if we apply the logic above to the "CMA-ES" method in Ant Morphology tasks, almost every method should be marked in bold. Therefore, we choose the method with the highest empirical mean except the "CMA-ES" method. Similar logic is also applied to the "PGS" in the superconductor task.
>
> Regarding the question of "why does it make sense to compare DEMO with the best unbolded method", we acutually treat all bolded methods have the competitive performance. Therefore, we test if the proposed DEMO is significantly better than other baselines.
>
> Best regards,
>
> Authors

---

### Review · Reviewer_4Lsc · 2025-02-21

**Summary Of Contributions:**

This paper presents a method to address the offline optimization problem in which an offline dataset of a black-box function is given and we want to find its maxima. A naive approach is to learn a surrogate of this function using the offline data and then perform gradient ascent on the input to this surrogate to find its maxima.

However, the main issue with this approach is that the surrogate might predict inaccurately as the gradient ascent process moves away from its offline data regime, which results in overestimation and suboptimal overall performance. The existing literature has provided numerous approaches to constraint the gradient ascent or search process in general such that it focuses more on the safe regime where the surrogate is reliable.

Under this context, this paper presents a design-editing method where top K maximizers of the surrogate are further edited via a diffusion process. The corresponding diffusion model seems to be learned to map between the offline data and Gaussian noises which however does not explain why it has the effect of mapping from base designs (maxima of surrogate) to designs with higher oracle value.

Overall, this is a relatively interesting idea but not well justified and also not well-positioned against diffusion-related works. The empirical comparison has also ignored a number of more recent works in offline optimization (see my comments below).

**Audience:**

Yes

**Claims And Evidence:**

No

**Requested Changes:**

Please revise the manuscript to address my concerns above in the weakness sections.

1. Most importantly, please provide a clear mathematical explanation to explain why does learning to map back and forth between Gaussian noises and offline data would have similar effect of mapping between low- and high-value designs during inference?

2. A positioning & comparison with https://openreview.net/pdf?id=K9Elg2JrvY is in my opinion important here because this prior work addresses the same problem using the same concept.

3. Please revise the related work section to include the missing references to provide a comprehensive review of literature.

**Strengths And Weaknesses:**

Strengths:

1. The paper is well-written.
2. The idea might be relatively new but the concept of design editing is not quite well-positioned with related work (see 2nd weakness)
3. The overall workflow is well-illustrated. I like figure 1 which is nice & informative.
4. The reported experiments appear strong.

Weaknesses:

First, it is unclear why the proposed idea work. From 4.3, the authors use the standard forward diffusion process to add Gaussian noise to the original input and then, learn a backward process to denoise it back.

As such, this is a vanilla diffusion prior mapping back and forth between the Gaussian noise distribution and the implicit offline data distribution. Why does it help map from the base designs to high-scoring design if it was not trained to do so?

Second, I believe this is also not the first work on design editing. For example, prior to this work, this has been explored here:
https://openreview.net/pdf?id=K9Elg2JrvY

The above work explicitly trained a generalized diffusion process to map between the distributions of low- and high-value designs. The authors should at least position with this work, i.e., why does learning to map back and forth between Gaussian noises and offline data would have similar effect of mapping between low- and high-value designs during inference?

Third, the authors do miss quite a number of more recent works (aside from the above) in their related work section. To point out a few:

1. ExPT: https://arxiv.org/abs/2310.19961 (NeurIPS-23)
2. Match-OPT: https://openreview.net/pdf?id=mv9beA1wDF (ICML-24)
3. BOSS: https://openreview.net/pdf?id=aLSA3JH08h (ICML-24)
4. aSCR: https://arxiv.org/abs/2402.06532 (NeurIPS-24)
5. IGNITE: https://openreview.net/pdf?id=ag7piyoyut (NeurIPS-24)

While I do understand that it might be hard to compare empirically with all existing works, it is still important to provide a thorough positioning and focus the baseline on most relevant work.

---

> ### Author Response · Authors · 2025-02-26
> **Reply to Reviewer 4Lsc**
>
> ## General Reply
> We thank the reviewer for the detailed and constructive feedback. We appreciate the critical evaluation of our approach and the insightful questions.
> We carefully address each point below and have updated our revised manuscript accordingly.
>
> ## Weaknesses and Requested Changes
> > First, it is unclear why the proposed idea work. From 4.3, the authors use the standard forward diffusion process to add Gaussian noise to the original input and then, learn a backward process to denoise it back. As such, this is a vanilla diffusion prior mapping back and forth between the Gaussian noise distribution and the implicit offline data distribution. Why does it help map from the base designs to high-scoring design if it was not trained to do so? Please provide a clear mathematical explanation to explain why does learning to map back and forth between Gaussian noises and offline data would have similar effect of mapping between low- and high-value designs during inference?
>
> Our method works because the diffusion model, although trained only to denoise Gaussian noise into realistic designs, effectively captures the structure of the valid design manifold.
> When pseudo candidates generated via gradient ascent drift away from this manifold, the diffusion-based editing process "pulls" them back to regions of high validity, thereby preserving the high predicted scores while correcting for extrapolation errors.
> One could interpret our approach as a single-phase Bayesian inference problem, where the diffusion model serves as a learned approximated prior distribution over the design space, and the likelihood can be approximated from the learned surrogate model.
> The optimization process can be seen as a form of approximate posterior inference—where our two-phase approach first approximates where high-score probability mass lies and then refines the candidates using the learned diffusion prior (for more details on this Bayesian perspective, please refer to Appendix A.6).
>
> | Gradient Ascent Step                                 | Ant   | TF8   |
> | ---------------------------------------------------- | ----- | ----- |
> | Predicted Score of Pseudo Design Candidates          | 0.950 | 2.916 |
> | Ground-truth Score of Pseudo Design Candidates       | 0.942 | 0.912 |
> | Ground-truth Score of Edited Final Design Candidates | 0.968 | 0.982 |
>
> Moreover, we run additional quantitative analysis. From the table above, we can observe that the pseudo design candidates usually have over optimistic predicted property scores but low ground-truth score.
> After the editing process, the ground-truth scores of the final candidates have been improved.
>
> > A positioning & comparison with https://openreview.net/pdf?id=K9Elg2JrvY is in my opinion important here because this prior work addresses the same problem using the same concept.
>
> We thank the reviewer for highlighting the work at https://openreview.net/pdf?id=K9Elg2JrvY. We would like to clarify that this work is a concurrent effort rather than a prior work, as our paper was submitted on January 10, 2025—prior to the public appearance of the referenced paper, which was under submission to ICLR and can be publicly accessed later on January 22, 2025.
>
> Nonetheless, we appreciate the opportunity to compare and position our approach relative to that work.
> We would like to the following paragraph to the section of related work and mark them in orange in our revised manuscript:
> "
> A concurrent work (Dao et al., 2025) unsurprisingly shares the core idea of leveraging diffusion processes to bridge the gap between low-and-high performing designs. This work explicitly trains a generalized diffusion process to map directly between the distributions of low- and high-value designs. In contrast, we leverage a diffusion prior in a two-phase process by first generating pseudo design candidates via surrogate-based gradient ascent, and then employing a diffusion model to edit these candidates so that they remain on the valid design manifold. Our decoupled two optimization phases and modularity provide enhanced stability and flexibility.
> "
>
>
> > Please revise the related work section to include the missing references to provide a comprehensive review of literature.
>
> We thank the reviewer for highlighting these recent works. In the revised manuscript, we have incorporated the suggested references and highlight them in orange—ExPT, Match-OPT, BOSS, aSCR, and IGNITE—and provided a thorough positioning of our method relative to these works. This expanded related work section offers a more comprehensive review of the literature and further clarifies how our two-phase approach in DEMO differs from and complements these existing methods.

---

> ### Author Response · Authors · 2025-03-04
> **Looking Forward to Your Feedback**
>
> Dear Reviewer 4Lsc,
>
> We sincerely appreciate your detailed and constructive feedback, which has helped us refine our manuscript. Your insights, particularly regarding the theoretical justification, quantitative analysis, and positioning relative to existing works, have been invaluable in improving the clarity and depth of our contributions.
>
> To address your concerns, we have carefully incorporated the following updates into our revised manuscript:
>
> - We explain the mathematical intuition behind our method and provide a mathematical explanation from Bayesian Inference perspective in the updated manuscript (Appendix A.6).
> - We present additional quantitative analysis demonstrating the effectiveness of the diffusion-based editing process in improving ground-truth scores of design candidates (Appendix A.5).
> - We clarify the positioning of our work relative to a concurrent submission (Dao et al., 2025) and highlight key differences in methodology and optimization strategy. This comparison has been added to the related work section and marked in orange.
> - We expand the related work section to include additional references (ExPT, Match-OPT, BOSS, aSCR, and IGNITE), ensuring a more comprehensive literature review.
>
> We greatly appreciate your time and consideration. We wanted to check if there are any additional concerns or clarifications needed from our side. We look forward to your response and further discussion.
>
> Best regards,
>
> Authors

---

> > ### Comment · Reviewer_4Lsc · 2025-03-12
> > **Quick follow-up**
> >
> > Dear Authors,
> >
> > Thank you for the detailed explanation & updates.
> >
> > I agree with the updates regarding the last two bullets (in your response). I also appreciate the additional results in the second bullet.
> >
> > But, I only partially agree with the explanation in the first bullet. Let's say we agree that the diffusion process captures the structure of the manifold & help pull the design back whenever it drifts away during gradient ascent.
> >
> > Your response suggests that whenever it is pulled back, it will be back to the high-value regime but I am not sure why it is guaranteed? What prevents it from mapping it back to the low-value regime?

---

> > > ### Author Response · Authors · 2025-03-12
> > > **Reply to the concern of Reviewer 4Lsc**
> > >
> > > Dear Reviewer 4Lsc,
> > >
> > > Thank you for your follow-up question and for your positive comments on our updates. We agree that our explanation should clarify that the diffusion editing will always guarantee to map a candidate back to the high-value regime. Instead, our approach is based on the following rationale:
> > >
> > > - **Trade-off Between Gradient Ascent and Diffusion Prior:**  In offline MBO, gradient ascent on the surrogate model suffers from the out-of-distribution (OOD) issue. On the other hand, relying solely on the diffusion prior—which is learned from the offline dataset that primarily contains relatively low-scoring designs—may cause the editing process to revert the candidate back to the low-value regime. Our method strikes a balance by using gradient ascent to push candidates toward higher predicted scores, while the diffusion-based editing serves as a regularization mechanism that pulls candidates back onto the valid data manifold. We achieve this trade-off through the amount of noise added during the perturbation of the pseudo design candidates. Rather than fully reverting a candidate to the low-value state, we choose an intermediate noise level (in our experiments, we use $m=600$ out of $M=1000$ for the maximum noise level) which ensures that the diffusion edit sufficiently regularizes the candidate—mitigating the OOD effect—while still preserving the gains achieved by gradient ascent. As shown in our analysis in Appendix A.2, selecting an excessively low $m$ causes the model to adhere too closely to the pseudo design candidates (thereby not fully correcting for OOD issues), whereas choosing an overly high $m$ biases the output toward the distribution of existing designs, neglecting the beneficial guidance provided by the gradient-based optimization. Moreover, the superior performance of our method in empirical experiments validates that our assumption about the OOD challenge is correct and that our approach effectively identifies and balances the trade-off betweem gradient ascent and diffusion prior. Last, we acknowledge that if $m$ is set too high, it may indeed cause the final design candidates overly conservative, but our experiments indicate that the chosen intermediate value of m provides an optimal trade-off.
> > >
> > > - **Bayesian Perspective:**  From a Bayesian viewpoint, our approach can be interpreted as performing an approximate posterior inference in two stages: first, the pseudo design candidates obtained via gradient ascent approximate the region of high posterior probability, and then the diffusion-based editing acts as a regularizer by incorporating the prior learned from the offline data. Although the diffusion model is not explicitly trained to map low-value designs to high-value ones, this Bayesian interpretation explains why the diffusion edit improves the final candidate designs.
> > >
> > > We hope this explanation clarifies our approach. Please let us know if this addresses your concerns or if there are any further aspects you would like us to elaborate on.
> > >
> > > Best regards,
> > >
> > > Authors

---

> > > > ### Comment · Reviewer_5gy3 · 2025-03-12
> > > > **I agree this is a weakness of the method**
> > > >
> > > > I agree with reviewer 4Lsc here. I think it would be helpful to emphasize that the performance of this method will depend on careful tuning to ensure it is not too OOD but also not too in-distribution.

---

> > > > > ### Author Response · Authors · 2025-03-12
> > > > > **Reply to Reviewer 4Lsc and 5gy3**
> > > > >
> > > > > Dear reviewer 4Lsc and 5gy3,
> > > > >
> > > > > To explicitly emphasize that the performance of our proposed DEMO method depends on careful tunings of hyperparameters to obtain competitive performance, we incorporate the following statements in the conclusion and discussion section (section 6), which are highlighted in orange color:
> > > > >
> > > > > > "One limitation of our proposed DEMO method is that its performance depends on the tuning of the hyperparameter $m$, which controls the amount of noise added during the design editing process, ensuring that the final design candidates are not extremely influenced by the extrapolation error of the surrogate while also avoiding an overly conservative edit that would revert them back to the low-scoring regime. It is important to note that such hyperparameter tuning for regularizations to balance overestimation and overconservatism is not unique to our approach; other methods in offline MBO may face similar challenges. Nonetheless, our experiments show that when $m$ is appropriately tuned, DEMO achieves a favorable trade-off and delivers competitive performance. We discuss additional limitations and potential negative impacts in Appendix A.7 and Appendix A.8, respectively."
> > > > >
> > > > > We hope this revision further clarify the our method and its limitations. Please let us know if this resolves your concerns or if there are any additional aspects you would like us to clarify further.
> > > > >
> > > > > Best regards,
> > > > >
> > > > > Authors

---

> > > > > > ### Author Response · Authors · 2025-03-17
> > > > > > **Follow up on the reply to Reviewer 4Lsc and 5gy3**
> > > > > >
> > > > > > Dear reviewer 4Lsc and 5gy3,
> > > > > >
> > > > > > We hope you are doing well. We are following up on our previous response regarding your suggestions, particularly on emphasizing the dependency of our proposed DEMO method on hyperparameter tuning. As mentioned, we have explicitly addressed this point by incorporating the clarification in the Conclusion and Discussion section (Section 6) of our revised manuscript. These revisions are highlighted in orange for easy reference.
> > > > > >
> > > > > > We hope this addition further clarifies our method and its limitations. Please let us know if this resolves your concerns or if there are any additional aspects you would like us to clarify further. We greatly appreciate your time and thoughtful feedback.
> > > > > >
> > > > > > Best regards,
> > > > > >
> > > > > > Authors

---

### Review · Reviewer_6yVc · 2025-02-21

**Summary Of Contributions:**

The paper introduces DEMO (Design Editing for Offline Model-based Optimization), which combines gradient-based optimization with diffusion models in a two-phase approach for offline model-based optimization. The first phase generates pseudo design candidates through gradient ascent on a surrogate model, while the second phase uses a diffusion prior to "edit" these candidates through a noise-and-denoise process to ensure they remain within the valid design distribution. The key insight driving this approach is that designs closer to the observed data manifold are more likely to be genuinely good rather than artifacts of model extrapolation.

**Audience:**

Yes

**Broader Impact Concerns:**

I don't see any potential concerns here.

**Claims And Evidence:**

Yes

**Requested Changes:**

1. Theoretical Framework: Is it possible to formulate DEMO as a single-phase Bayesian inference problem with diffusion as prior and gradient descent on posterior given y.

2. Quantitative Analysis: What's the difference of expected scores and oracle scores between pseudo-candidates and diffusion-edited designs? This would better justify the two-phase approach.

3. Convergence Study: Examine how gradient descent convergence in first phase affects diffusion editing benefits. Does editing remain helpful with very long gradient descent runs (i.e. very far away from training distributions)?

**Strengths And Weaknesses:**

Strengths:

From a technical perspective, the paper presents an approach that effectively combines gradient-based optimization with diffusion models, providing a way to address the out-of-distribution problem. The theoretical justification for staying near the data manifold is well-reasoned and clearly explained. The empirical validation is particularly strong, with comprehensive experiments across seven different tasks, thorough ablation studies, and statistical significance tests demonstrating meaningful improvements. The method achieves impressive performance metrics. The paper is also well-written with clear motivation, problem setup, and effective visualizations that help explain the method.

---

> ### Author Response · Authors · 2025-02-26
> **Reply to Reviewer 6yVc**
>
> ## General Reply
> We thank the reviewer for the thorough and encouraging evaluation of our work. We are pleased that our method is both technically sound and practically effective. We also appreciate your valuable suggestions regarding the theoretical formulation, quantitative comparisons, and convergence analysis. We carefully address these points below and merge them to our revised manuscript. Thank you again for your constructive feedback.
>
> ## Requested Changes
> > Theoretical Framework: Is it possible to formulate DEMO as a single-phase Bayesian inference problem with diffusion as prior and gradient descent on posterior given y.
>
> We appreciate your suggestion to explore a unified Bayesian formulation of DEMO. One could interpret our approach as a single-phase Bayesian inference problem, where the diffusion model serves as a learned prior over the design space. While this interpretation is conceptually appealing, it also poses challenges in terms of likelihood modeling, accuracy of the prior modeling, and effectiveness of the regularization. We include a discussion of this Bayesian perspective in our revised manuscript in Appendix A.6 and mark it in red, highlighting both its theoretical merits and the practical advantages of our current modular two-phase approach. Also, we add the following sentence in the conclusion section and highlight it with red color: "We further discuss the connection between our method and Bayesian inference in Appendix A.6".
>
> In short, from a Bayesian perspective, our objective is to find the design $\boldsymbol{x}$ that maximizes the posterior probability:
> $p(\boldsymbol{x} | y) \propto p(y | \boldsymbol{x}) p(\boldsymbol{x}),$ where $p(y|\boldsymbol{x})$ is the likelihood of obtaining a property score $y$ given a design $\boldsymbol{x}$, and $p(\boldsymbol{x})$ is the prior distribution over designs learned from the offline dataset. Taking the logarithm, we have: $\log p(\boldsymbol{x}|y) = \log p(y|\boldsymbol{x}) + \log p(\boldsymbol{x})+C,$ with $C$ being a constant that does not affect the optimization. To find the design that maximizes the posterior, we aim to solve: $\boldsymbol{x}^* = \arg\max_{\boldsymbol{x} \in \mathcal{X}} [\log p(y|\boldsymbol{x}) + \log p(\boldsymbol{x})].$ In the offline model-based optimization (MBO), we typically do not have a direct expression for $p(y|\boldsymbol{x})$.
> By following [r1, r2], we can model the probability density using the Boltzmann distribution and the learned surrogate as follows:
> $p(y|\boldsymbol{x}) \approx p_{\boldsymbol{\theta}}(y|\boldsymbol{x}) = \frac{e^{\gamma f_{\boldsymbol{\theta}}(\boldsymbol{x})}}{Z},$
> where $\gamma$ is the scaling factor, and $Z$ is the normalization constant. Taking the logarithm of this expression, we obtain: $\log p(y|\boldsymbol{x}) \approx \gamma f_{\boldsymbol{\theta}}(\boldsymbol{x}) - \log Z.$ To update the design, we can leverage the surrogate model to approximate the gradient of $\log p(y|\boldsymbol{x})$ and the diffusion prior to approximate the gradient of $\log p(\boldsymbol{x})$.
>
>
> [r1] Lee, Seul, et al. “Exploring Chemical Space with Score-Based Out-of-Distribution Generation.” (ICML)
>
> [r2] Yuan, Ye, et al. “ParetoFlow: Guided Flows in Multi-Objective Optimization.” (ICLR)

---

> ### Author Response · Authors · 2025-02-26
> **Additional Reply to Reviewer 6yVc**
>
> **Ant**
>
> | Gradient Ascent Step                                 | 100   | 200   | 300   |
> | ---------------------------------------------------- | ----- | ----- | ----- |
> | Predicted Score of Pseudo Design Candidates          | 0.950 | 1.210 | 1.340 |
> | Ground-truth Score of Pseudo Design Candidates       | 0.942 | 0.887 | 0.845 |
> | Ground-truth Score of Edited Final Design Candidates | 0.968 | 0.935 | 0.928 |
>
> **TF8**
>
> | Gradient Ascent Step                                 | 100   | 200   | 300   |
> | ---------------------------------------------------- | ----- | ----- | ----- |
> | Predicted Score of Pseudo Design Candidates          | 2.916 | 5.570 | 8.311 |
> | Ground-truth Score of Pseudo Design Candidates       | 0.912 | 0.895 | 0.895 |
> | Ground-truth Score of Edited Final Design Candidates | 0.982 | 0.958 | 0.914 |
>
>
>
>
>
>
>
> > Quantitative Analysis: What's the difference of expected scores and oracle scores between pseudo-candidates and diffusion-edited designs? This would better justify the two-phase approach.
>
> We run additional experiments on one continuous task Ant and one discrete task TF8.
> As shown in the table above, the pseudo design candidates usually have over optimistic predicted score, but their ground-truth scores evaluated by the oracle are not such high.
> After editing the pseudo design candidates by our approach, the ground-truth scores are effectively increased.
>
> > Convergence Study: Examine how gradient descent convergence in first phase affects diffusion editing benefits. Does editing remain helpful with very long gradient descent runs (i.e. very far away from training distributions)?
>
> Similarly, we run extended experiments on one continuous task Ant and one discrete task TF8.
> In our original setting, we run 100 gradient ascent steps to acquire the pseudo design candidates.
> 200 and 300 gradient ascent steps are considered in the additional experiments, where we can mimic the distribution very far away from the training distribution.
> As we can see from the tables above, even though the performance degrades as the number of gradient ascent steps increases, our proposed approach is still helpful for editing the pseudo design candidates and thus improving the quality of the final candidates.
>
> We add the following sentences to the end of experiment results subsection and highlight them in red: "Additional quantitative analysis of the predicted and ground-truth property scores of the pseudo design candidates and final candidates is provided in Appendix A.5. We also study the influence of the number of gradient ascent steps in the convergence study detailed in Appendix A.5."
> The discussion of quantitative analysis and convergence study are included in Appendix A.5.
> Please check our revised manuscript.

---

> ### Author Response · Authors · 2025-03-04
> **Looking Forward to Your Feedback**
>
> Dear Reviewer 6yVc,
>
> We sincerely appreciate your thorough review and valuable feedback on our work. Your insights have significantly helped us refine our theoretical formulation, quantitative comparisons, and convergence analysis.
>
> To address your suggestions, we have carefully incorporated the following updates into our revised manuscript:
>
> - We discuss the Bayesian perspective of our method in Appendix A.6.
> - We provide additional quantitative analysis on the predicted vs. ground-truth property scores of pseudo design candidates and diffusion-edited designs (Appendix A.5).
> - We extend the convergence study by analyzing the impact of longer gradient ascent steps on diffusion editing (Appendix A.5).
> - We update the manuscript with clarifications and highlight the relevant changes in red for easy reference.
>
> We would greatly appreciate any further comments or suggestions you may have. We kindly follow up to inquire if there are any additional concerns that require our attention. We look forward to your response.
>
> Best regards,
>
> Authors

---

> > ### Author Response · Authors · 2025-03-17
> > **Follow up on the previous reply**
> >
> > Dear Reviewer 6yVc,
> >
> > We hope you are doing well. We wanted to follow up on our previous response regarding your insightful suggestions on our theoretical formulation, quantitative comparisons, and convergence analysis.
> >
> > As mentioned, we have carefully incorporated the updates in our revised manuscript. We would greatly appreciate any further comments or suggestions you may have. Please let us know if there are any additional concerns that require our attention. We look forward to your response.
> >
> > Best regards,
> >
> > Authors

---

### Review · Reviewer_YKnZ · 2025-03-08

**Summary Of Contributions:**

The paper presents Design Editing for Offline Model-based Optimization (DEMO), a novel method to improve offline model-based optimization (MBO) using gradient ascent on a surrogate model to generate pseudo design candidates. The model is able to address out-of-distribution (OOD) issues by refining pseudo design candidates through diffusion-based editing. Empirical validation across multiple MBO tasks show performance improvements over existing optimization methods.

**Audience:**

Yes

**Claims And Evidence:**

Yes

**Requested Changes:**

While the results show that DEMO outperforms baselines, there is limited discussion on computational efficiency and scalability. Since diffusion models are typically more expensive than traditional optimization approaches, an analysis of runtime, memory usage, and efficiency trade-offs would help assess its practicality for large-scale problems.

**Strengths And Weaknesses:**

The paper identifies out-of-distribution errors as a key challenge in offline MBO and explains how surrogate models often extrapolate unreliably when optimizing outside the training data distribution. The idea of using a diffusion prior to project pseudo design candidates back into the valid distribution is a conceptually sound and novel approach to mitigating these issues. The paper also includes a thorough empirical evaluation across seven diverse MBO tasks, covering both continuous and discrete domains. The paper is well-written and easy to understand. The figures are also clear and informative.

---

> ### Author Response · Authors · 2025-03-10
> **Reply to Reviewer YKnZ**
>
> Dear Reviewer YKnZ,
>
> Thank you for your thorough and insightful review. We appreciate your recognition of the conceptual soundness and empirical validation of DEMO, as well as your constructive suggestion to include an analysis of computational efficiency and scalability.
>
> In our revised manuscript, we have added a dedicated subsection in the Experiments section titled “Computational Efficiency Analysis,” which reports GPU memory usage, training and optimization runtimes for each task, and provides a detailed comparison with another diffusion-based method (DDOM) as well as representative surrogate-based methods (Mean Ensemble and Tri-mentoring). Our results show that DEMO is not only more effective—achieving superior mean rankings—but also more efficient than DDOM in optimizing design candidates, while still maintaining competitive total training time and optimization time efficiency compared to traditional surrogate-based approaches. Please check our revised manuscript Section5.7 for more details, which is already highlighted in pink color.
>
> We hope these additions address your concerns regarding efficiency and scalability, demonstrating that DEMO offers a practical solution for real-world applications. Thank you again for your valuable feedback.
>
> Best regards,
>
> Authors

---

> > ### Author Response · Authors · 2025-03-17
> > **Follow up on the previous reply**
> >
> > Dear Reviewer YKnZ,
> >
> > We hope this message finds you well. We wanted to follow up on our previous response regarding your insightful suggestions on computational efficiency and scalability in DEMO.
> >
> > As mentioned, we have incorporated a dedicated subsection in our revised manuscript (Section 5.7: Computational Efficiency Analysis), where we present GPU memory usage, training and optimization runtimes, and a comparative analysis with both diffusion-based (DDOM) and surrogate-based methods (Mean Ensemble and Tri-mentoring). These updates are highlighted in pink for easy reference.
> >
> > We would greatly appreciate any further thoughts or feedback you may have on these revisions. Please let us know if there are any remaining concerns or aspects that require further clarification.
> >
> > Thank you again for your time and valuable insights. We truly appreciate your efforts in reviewing our work.
> >
> > Best regards,
> >
> > Authors

---

### Decision · Action_Editor_Cw2z · 2025-04-14

**Recommendation:** Accept as is

**Comment:**

The reviewers initially raised concerns about statistical methodology, performance claims, theoretical justification, and hyperparameter sensitivity, but ultimately recommended acceptance after the authors comprehensively addressed each issue by implementing proper statistical testing with Bonferroni corrections, revising performance claims from "state-of-the-art" to "competitive," adding Bayesian theoretical framing, and acknowledging limitations regarding hyperparameter tuning. The experimental validation across seven diverse MBO tasks demonstrates the method's effectiveness and generalizability, making it relevant to researchers in optimization, generative models, and their applications.

TMLR's audience would find value in this paper on Design Editing for Offline Model-based Optimization, which addresses the out-of-distribution challenge in offline MBO through a novel two-phase approach combining gradient-based optimization with diffusion models.

**Audience:**

Yes, TMLR's audience would likely be interested in the findings of this paper. The work addresses an important challenge in offline model-based optimization by introducing a novel approach (DEMO) that combines gradient-based optimization with diffusion models to balance between finding high-scoring designs and ensuring they remain valid. Machine learning researchers working on optimization, generative models, and their applications would find value in this methodology. The paper's empirical validation across diverse tasks and its theoretical framing through a Bayesian perspective would appeal to both practitioners and theoreticians in the TMLR community.

**Claims And Evidence:**

The claims made in the submission are supported by accurate, convincing, and clear evidence through comprehensive experiments across seven offline MBO tasks with appropriate statistical testing. Initially, there were concerns about statistical methodology and performance claims, but the authors addressed these by implementing proper Bonferroni corrections, revising claims from "state-of-the-art" to "competitive," and enhancing theoretical justification with a Bayesian perspective. They acknowledged the method's dependency on hyperparameter tuning (particularly the noise level parameter) and added analysis of computational efficiency and scalability. The evidence is sufficient to support the revised claims, with the authors being transparent about limitations, which strengthens the paper's scientific contribution.